# Incentive-Aware Synthetic Control: Accurate Counterfactual Estimation via Incentivized Exploration

**Daniel Ngo**[*][†]                                    *tuandung.ngo@jpmchase.com*
*J.P. Morgan Chase AI Research*

**Keegan Harris**[*]                                    *keegan.harris@berkeley.edu*
*University of California, Berkeley*

**Anish Agarwal**                                    *aa5194@columbia.edu*
*Columbia University*

**Vasilis Syrgkanis**                                    *vsyrgk@stanford.edu*
*Stanford University*

**Zhiwei Steven Wu**                                    *zstevenwu@cmu.edu*
*Carnegie Mellon University*

**Reviewed on OpenReview:** *https://openreview.net/forum?id=koln3ufP5c*

## Abstract

*Synthetic control methods (SCMs)* are a canonical approach used to estimate treatment effects from panel data in the internet economy. We shed light on a frequently overlooked but ubiquitous assumption made in SCMs of "overlap": a treated unit can be written as some combination—typically, convex or linear—of the units that remain under control. We show that if units select their own interventions, and there is sufficiently large heterogeneity between units that prefer different interventions, overlap will not hold. We address this issue by proposing a recommender system which *incentivizes* units with different preferences to take interventions they would not normally consider. Specifically, leveraging tools from information design and online learning, we propose an SCM that incentivizes exploration in panel data settings by providing incentive-compatible intervention recommendations to units. We establish this estimator obtains valid counterfactual estimates without the need for an *a priori* overlap assumption. We extend our results to the setting of synthetic interventions, where the goal is to produce counterfactual outcomes under all interventions, not just control. Finally, we provide two hypothesis tests for determining whether unit overlap holds for a given panel dataset.

## 1 Introduction

A ubiquitous task in the internet economy is to estimate counterfactual outcomes for a group of *units* (e.g. people, products, geographic regions) under different *interventions* (e.g. marketing campaigns, discount levels, legal regulations) over *time*. Such multi-dimensional data are often referred to as *panel data*, where the different units may be thought of as rows of a matrix and the time-steps as columns. A prominent framework for counterfactual inference using panel data is that of *synthetic control methods* (SCMs) Abadie & Gardeazabal (2003); Abadie et al. (2010), which aim to estimate the counterfactual outcome under *control*, i.e. no treatment, for units that were treated.

---

[*]Equal contribution.
[†]Work was done during a visit at CMU.

In the SCM framework[1], there is a *pre-intervention* time period, during which all units are under *control* (i.e. under no intervention). After the pre-intervention period there is a *post-intervention* time period, where each unit is given exactly one intervention from a set of possible interventions (which includes control).

The goal of SCMs is to estimate the counterfactual outcomes under control for a treated unit during the post-intervention period. To do this, SCMs broadly follow two steps:

1. during the pre-intervention period, linearly regress the outcomes of a treated unit against the outcomes of the units that remain under control, i.e. the *donor* units;

2. use the learned linear model and the outcomes of the donor units during the post-intervention period to produce a counterfactual trajectory of the treated unit under control.

While SCMs were originally applied to public policy domains (e.g. Abadie et al. (2010); Freire (2018)), they are now often used to estimate counterfactuals in various internet domains, ranging from surge pricing for ride-sharing applications Uber (2019) to the effectiveness of advertisements across different geographic regions EBay (2022).

From the description of SCMs laid out above, it is clear that for the counterfactual estimate to be valid, the treated unit's potential outcomes under control should be linearly expressible by the observed outcomes of the donor units.[2] To provide statistical guarantees on the performance of SCMs, such intuition has traditionally been made formal by making an *overlap* assumption on the relationship between the outcomes of the donor units and the test unit; for instance of the following form:

**Unit Overlap Assumption.** *Denote the potential outcome for unit $i$ under intervention $d$ at time $t$ by $y_{i,t}^{(d)} \in \mathbb{R}$. For a given unit $i$ and intervention $d$, there exists a set of weights $\omega^{(i,d)} \in \mathbb{R}^{N_d}$ such that*

$$\mathbb{E}[y_{i,t}^{(d)}] = \sum_{j \in [\![N_d]\!]} \omega^{(i,d)}[j] \cdot \mathbb{E}[y_{j,t}^{(d)}] \text{ for all } t \in [\![T]\!],$$

*where $T$ is the number of time-steps, $N_d$ is the number of donor units who have received intervention $d$. The expectation is taken with respect to the randomness in the unit outcomes.*

More broadly, previous works that provide theoretical guarantees for SCMs assume there exists some underlying mapping $\omega^{(i,d)}$ (e.g. linear or convex) through which the outcomes of a treated unit (unit $i$) may be expressed by the outcomes of the $N_d$ donor units. Since such a condition appears to be necessary in order to make valid counterfactual inference, assumptions of this nature are ubiquitous when proving statistical guarantees about SCMs (e.g. Abadie et al. (2010); Amjad et al. (2018); Agarwal et al. (2020b; 2023)).[3]

However, despite their ubiquity, such overlap assumptions may not hold in all domains in which one would like to apply SCMs. For example, consider a streaming service with two service plans (i.e. interventions): a yearly subscription (the treatment) and a pay-as-you-go model (the control). Suppose that all streamers (the units in this example) are initially signed up for the pay-as-you-go model, and may choose to switch to the subscription model after a trial period (the pre-intervention period). Furthermore, suppose that the streaming service wants to determine the effectiveness of its subscription program on user engagement (the unit outcome of interest) over the next year (the post-intervention period).

Under this setting, the subpopulation of streamers who self-select the subscription plan is most likely those who believe they will consume large amounts of content on the platform. In contrast, those who pay-as-they-go most likely believe they will consume less content. The two subpopulations may have very different experiences under the two business plans (due to their differing viewing habits), leading to two different sets of potential outcomes that may not overlap, i.e., are not linearly expressible in terms of each other. This could make it difficult to draw conclusions about the counterfactual user engagement levels of one subpopulation using the realized engagement levels of the other.

---

[1]See Section 2.1 for a more in depth description of the SCM framework we consider.

[2]See Section 3 for a mathematical justification of this fact.

[3]We discuss the role of the Unit Overlap Assumption and related conditions in the literature in Section 1.1.

In this work, our goal is to leverage tools from information design to incentivize the exploration of different treatments by non-overlapping unit subpopulations in order to obtain valid counterfactual estimates using SCMs. Specifically, we adopt tools and techniques from the literature on incentivizing exploration in multi-armed bandits (e.g. Mansour et al. (2020); Slivkins (2024)) to show how the principal—the platform running the SCM—can leverage knowledge gained from previous interactions with similar units to persuade the current unit to take an intervention they would not normally take. The principal achieves this by sending a *signal* or *recommendation* to the unit with information about which intervention is best for them. The principal's *recommendation policy* is designed to be *incentive-compatible*, i.e. it is in the unit's best interests to follow the intervention recommended to them by the policy. In our streaming service example, incentive-compatible signalling may correspond to recommending a service plan for each user based on their usage of the platform in a way that guarantees that units are better off in expectation when purchasing the recommended service plan. Our procedure ensures that the Unit Overlap Assumption becomes satisfied over time, which enables the principal to do valid counterfactual inference using off-the-shelf SCMs after they have interacted with sufficiently many units.

**Overview of paper.** We introduce an online learning model where the principal observes panel data from a population of units with differing preferences. We show that the Unit Overlap Assumption is indeed a necessary condition to obtain valid counterfactual estimates in our setting. To achieve overlap, we introduce a recommender system for incentivizing exploration when there are two interventions: treatment and control. At a high level, we adapt the "hidden exploration" paradigm from incentivized exploration in bandits to the panel data setting: First, randomly divide units into "exploit" units and "explore" units. For all exploit units, recommend the intervention that maximizes their estimated expected utility, given the data seen so far. For every explore unit, recommend the intervention for which the principal would like the Unit Overlap Assumption to be satisfied. Units are not aware of their exploit/explore designation by the algorithm, and the explore probability is chosen to be low enough such that the units have an incentive to follow the principal's recommendations. After recommendations have been made to sufficiently-many units, we show that with high probability, the Unit Overlap Assumption is satisfied for all units and either intervention of our choosing. This enables the use of existing SCMs to obtain finite sample guarantees for counterfactual estimation for all units under control after running Algorithm 1.

We extend our methods to the setting of synthetic interventions Agarwal et al. (2020b), in which the principal may wish to estimate counterfactual outcomes under different treatments in addition to control (Appendix D). We also provide two hypothesis tests for checking whether the Unit Overlap Assumption holds for a given treated unit and set of donor units. Finally, we empirically evaluate the performance of our recommender system. We find that it enables consistent counterfactual outcome estimation when the Unit Overlap Assumption does not hold *a priori* while existing SCMs alone do not.

## 1.1 Related work

**Causal inference and synthetic control methods.** Popular methods for counterfactual inference using panel data include SCMs Abadie & Gardeazabal (2003); Abadie et al. (2010), difference-in-differences Angrist & Pischke (2009); Ashenfelter & Card (1984); Bertrand et al. (2004), and clustering-based methods Zhang et al. (2019); Dwivedi et al. (2022). Within the literature on synthetic control, our work builds off of the line of work on *robust* synthetic control Amjad et al. (2018; 2019); Agarwal et al. (2020a;b; 2023), which assumes outcomes are generated via a *latent factor model* (e.g. Chamberlain (1984); Liang & Zeger (1986); Arellano & Honore (2000)) and leverages *principal component regression* (PCR) Jolliffe (1982); Massy (1965); Agarwal et al. (2019; 2020a) to estimate unit counterfactual outcomes. Our work falls in the small-but-growing line of work at the intersection of SCMs and online learning Chen (2023); Farias et al. (2022); Agarwal et al. (2023), although we are the first to consider unit incentives in this setting. Particularly relevant to our work is the model used by Agarwal et al. (2023), which extends the finite sample guarantees from PCR in the panel data setting to online settings. While Harris et al. (2024) also consider incentives in SCMs (albeit in an offline setting), they only allow for a principal who can *assign* interventions to units (e.g. can force compliance). As a result, the strategizing they consider is that of units who modify their pre-intervention outcomes in order to be assigned a more desirable intervention. In

---

**Protocol: Incentivizing Exploration for Synthetic Control**

For each unit $i \in [\![n]\!]$:

1. Principal observes pre-intervention outcomes $\mathbf{y}_{i,pre}$

2. Principal recommends an intervention $\widehat{d}_i \in \{0,1\}$

3. Unit $i$ chooses intervention $d_i \in \{0,1\}$

4. Principal observes post-intervention outcomes $\mathbf{y}_{i,post}^{(d_i)}$

---

Figure 1: Summary of our setting.

contrast, we consider a principal who cannot assign interventions to units but instead must *persuade* units to take different interventions by providing them with incentive-compatible recommendations.

**Unit overlap and similar conditions.** Our notion of unit overlap comes directly from the literature on robust synthetic control (see the previous paragraph for references). However, it is important to note that analogous assumptions appear more broadly in the literature on synthetic control. For example, Abadie et al. (2010) requires that the treated unit's pre-treatment outcomes and covariates lie in the convex hull of the donor units pool. Beyond synthetic control, similar assumptions to the Unit Overlap Assumption are also prevalent in the literature on other *matching*-based estimators such as *difference-in-differences* (Donald & Lang, 2007) or *clustering*-based methods (Zhang et al., 2019). Finally, it is worth noting that in causal inference, the term "overlap" is also commonly used to refer to propensity score overlap, which assumes that the probability a unit receives treatment is bounded in $(0,1)$. This is in contrast to the unit outcome/latent factor overlap that we consider.

**Incentivized exploration.** Our work draws on techniques from the growing literature on incentivizing exploration (IE) (Kremer et al., 2013; Mansour et al., 2020; 2022; Sellke & Slivkins, 2023; Immorlica et al., 2020; Sellke, 2023; Hu et al., 2022; Ngo et al., 2021).In IE, a principal interacts with a sequence of myopic agents over time. Each agent takes an action, and the principal can observe the outcome of each action. While each individual agent would prefer to take the action which appears to be utility-maximizing (i.e. to exploit), the principal would like to incentivize agents to explore different actions in order to benefit the population in aggregate. Motivation for IE includes online rating platforms and recommendation systems which rely on information collected by users to provide informed recommendations about various products and services.

## 2 Preliminaries

**Notation.** Subscripts are used to index the unit and time-step, while superscripts are reserved for interventions. We use $i$ to index units, $t$ to index time-steps, and $d$ to index interventions. For $x \in \mathbb{N}$, we use the shorthand $[\![x]\!] := \{1, 2, \ldots, x\}$ and $[\![x]\!]_0 := \{0, 1, \ldots, x-1\}$. $\mathbf{y}[i]$ denotes the $i$-th component of vector $\mathbf{y}$, where indexing starts at 1. We sometimes use the shorthand $T_1 := T - T_0$ for $T, T_0 \in \mathbb{N}_{>0}$ such that $T > T_0$. $\Delta(\mathcal{X})$ denotes the set of possible probability distributions over $\mathcal{X}$. $\mathbf{0}_d$ is shorthand for the vector $[0, 0, \ldots, 0] \in \mathbb{R}^d$. Finally, we use the notation $a \wedge b := \min\{a, b\}$ and $a \vee b := \max\{a, b\}$.

### 2.1 Our panel data setting

We consider an online setting in which the principal interacts with a sequence of $n$ units one-by-one for $T$ time steps each. As is standard in the literature on synthetic control, we posit that there is a pre-intervention period of $T_0$ time-steps, for which each unit $i \in [\![n]\!]$ is under the same intervention, i.e., under *control*. After the pre-intervention period, the principal either recommends the *treatment* to the unit or suggests that they remain under control (possibly using knowledge of the outcomes for units $j < i$). We denote the principal's *recommended* intervention to unit $i$ by $\widehat{d}_i \in \{0,1\}$, where 1 denotes the treatment and 0 the

control. After receiving the recommendation, unit $i$ chooses an intervention $d_i \in \{0, 1\}$ and remains under intervention $d_i$ for the remaining $T - T_0$ time-steps.[4] We use $\mathbf{y}_{i,pre} := [y_{i,1}^{(0)}, \ldots, y_{i,T_0}^{(0)}]^\top \in \mathbb{R}^{T_0}$ to denote unit $i$'s pre-treatment outcomes under control, and $\mathbf{y}_{i,post}^{(d)} := [y_{i,T_0+1}^{(d)}, \ldots, y_{i,T}^{(d)}]^\top \in \mathbb{R}^{T-T_0}$ to refer to unit $i$'s post-intervention potential outcomes under intervention $d$. We denote the set of possible pre-treatment outcomes by $\mathcal{Y}_{pre}$. See Figure 1 for a summary of our setting.

In order to impose structure on how unit outcomes are related for different units, time-steps, and interventions, we assume that outcomes are generated via the following *latent factor model*, a popular assumption in the literature on synthetic control (see references in Section 1.1).

**Assumption 2.1** (Latent Factor Model). *Suppose the outcome for unit $i$ at time $t$ under treatment $d \in \{0, 1\}$ takes the factorized form $\mathbb{E}[y_{i,t}^{(d)}] = \langle \mathbf{u}_t^{(d)}, \mathbf{v}_i \rangle$, $y_{i,t}^{(d)} = \mathbb{E}[y_{i,t}^{(d)}] + \varepsilon_{i,t}^{(d)}$, where $\mathbf{u}_t^{(d)} \in \mathbb{R}^r$ is a latent vector which depends only on the time-step $t$ and intervention $d$, $\mathbf{v}_i \in \mathbb{R}^r$ is a latent vector which only depends on unit $i$, and $\varepsilon_{i,t}^{(d)}$ is zero-mean sub-Gaussian random noise with variance at most $\sigma^2$. For simplicity, we assume that $|\mathbb{E}[y_{i,t}^{(d)}]| \leq 1, \ \forall \ i \in [\![n]\!], t \in [\![T]\!], d \in \{0, 1\}$.*

While we assume the existence of such structure in the data, we do *not* assume that the principal knows or gets to observe $\mathbf{u}_t^{(d)}$, $\mathbf{v}_i$, or $\varepsilon_{i,t}^{(d)}$. In the literature on synthetic control, the goal of the principal is to estimate unit-specific counterfactual outcomes under different interventions. Specifically, our target causal parameter is the (counterfactual) average expected post-intervention outcome.[5]

**Definition 2.2.** *(Average expected post-intervention outcome) The average expected post-intervention outcome of unit $i$ under intervention $d$ is $\mathbb{E}[\bar{y}_{i,post}^{(d)}] := \frac{1}{T_1} \sum_{t=T_0+1}^{T} \mathbb{E}[y_{i,t}^{(d)}]$, where the expectation is taken with respect to $(\epsilon_{i,t}^{(d)})_{T_0 < t \leq T}$.*

## 2.2 Recommendations and beliefs

The data generated by the interaction between the principal and a unit $i$ may be characterized by the tuple $(\mathbf{y}_{i,pre}, \widehat{d}_i, d_i, \mathbf{y}_{i,post}^{(d_i)})$, where $\mathbf{y}_{i,pre}$ are unit $i$'s pre-treatment outcomes, $\widehat{d}_i$ is the intervention recommended to unit $i$, $d_i$ is the intervention taken by unit $i$, and $\mathbf{y}_{i,post}^{(d_i)}$ are unit $i$'s post-intervention outcomes under intervention $d_i$.

**Definition 2.3** (Interaction History). *The interaction history at unit $i$ is the sequence of outcomes, recommendations, and interventions for all units $j \in [\![i-1]\!]$. Formally, $H_i := \{(\mathbf{y}_{j,pre}, \widehat{d}_j, d_j, \mathbf{y}_{j,post}^{(d_j)})\}_{j=1}^{i-1}$. We denote the set of all possible histories at unit $i$ as $\mathcal{H}_i$.*

We refer to the way the principal assigns interventions to units as a *recommendation policy*.

**Definition 2.4** (Recommendation Policy). *A recommendation policy $\pi_i : \mathcal{H}_i \times \mathcal{Y}_{pre} \rightarrow \Delta(\{0, 1\})$ is a (possibly stochastic) mapping from histories and pre-treatment outcomes to interventions.*

We assume that before the first unit arrives, the principal commits to a method for computing the sequence of recommendation policies $\{\pi_i\}_{i=1}^n$ which is fully known to all units. Whenever $\pi_i$ is clear from the context, we use the shorthand $\widehat{d}_i = \pi_i(H_i, \mathbf{y}_{i,pre})$ to denote the recommendation of policy $\pi_i$ to unit $i$.

In addition to having a corresponding latent factor, each unit has an associated *belief* over the effectiveness of each intervention. This is made formal through the following definitions.

**Definition 2.5** (Unit Belief). *Unit $i$ has prior belief $\mathcal{P}_{\mathbf{v}_i}$, which is a joint distribution over potential post-intervention outcomes $\{\mathbb{E}[\bar{\mathbf{y}}_{i,post}^{(0)}], \mathbb{E}[\bar{\mathbf{y}}_{i,post}^{(1)}]\}$. We use the shorthand $\mu_{v_i}^{(d)} := \mathbb{E}_{\mathcal{P}_{\mathbf{v}_i}}[\bar{y}_{i,post}^{(d)}]$ to refer to a unit's expected average post-intervention outcome, with respect to their prior $\mathcal{P}_{\mathbf{v}_i}$.*

**Definition 2.6** (Unit Type). *Unit $i$ is of type $\tau \in \{0, 1\}$ if $\tau = \arg\max_{d \in \{0,1\}} \mu_{v_i}^{(d)}$. We denote the set of all units of type $\tau$ as $\mathcal{T}^{(\tau)}$ and the dimension of the latent subspace spanned by type $\tau$ units as $r_\tau := \text{span}(\mathbf{v}_j : j \in \mathcal{T}^{(\tau)})$.*

---

[4]We envision a scenario where both interventions are available to units and they can choose between them.

[5]Our results hold in the more general setting where the target causal parameter is *any linear* function of the vector of expected post-intervention outcomes. We focus on the average as it is a common quantity of interest.

In other words, a unit's type is the intervention they would take according to their prior without any additional information about the effectiveness of each intervention. We consider the setting in which the possible latent factors associated with each type lie in *mutually orthogonal* subspaces (i.e. the Unit Overlap Assumption is *not* satisfied).See Section 3 for a simple theoretical example of such a data-generating process and Section 6 for an empirical example. If the principal could get the Unit Overlap Assumption to be satisfied, they could use existing synthetic control methods off-the-shelf to obtain valid finite sample guarantees.

As is standard in the literature on IE, we assume that units are Bayesian-rational and each unit knows its place in the sequence of $n$ units (i.e., their index $i \in [\![n]\!]$).Thus given recommendation $\widehat{d}_i$, unit $i$ selects their intervention $d_i$ such that $d_i \in \arg\max_{d \in \{0,1\}} \mathbb{E}_{\mathcal{P}_{\mathbf{v}_i}}[\bar{y}_{i,post}^{(d)}|\widehat{d}_i]$, i.e. they select the intervention $d_i$ which maximizes their utility in expectation over their prior $\mathcal{P}_{\mathbf{v}_i}$, conditioned on receiving recommendation $\widehat{d}_i$ from the principal.

**Definition 2.7** (Bayesian Incentive-Compatibility)**.** *We say that a recommendation $d$ is* Bayesian incentive-compatible *(BIC) for unit $i$ if, conditional on receiving intervention recommendation $\widehat{d}_i = d$, unit $i$'s average expected post-intervention outcome under intervention $d$ is at least as large as their average expected post-intervention outcome under any other intervention. Formally, $\mathbb{E}_{\mathcal{P}_{\mathbf{v}_i}}[\bar{y}_{i,post}^{(d)} - \bar{y}_{i,post}^{(d')}|\widehat{d}_i = d] \geq 0$ for every $d' \in \{0,1\}$. A recommendation policy is BIC if the above condition holds for every intervention which is recommended with non-zero probability.*

It is worth clarifying the behavioral interpretation of Definition 2.7. Formally, Definition 2.7 assumes that units are Bayesian-rational, understand the recommendation algorithm, and correctly condition on the information conveyed by receiving a particular recommendation. This assumption provides a clean and standard benchmark that ensures recommendations are followed and allows us to focus on the statistical question of inducing overlap. At the same time, we do not view this assumption as requiring literal Bayesian reasoning by agents in practice. In many real-world systems, users may instead follow recommendations with high but imperfect probability, rely on coarse beliefs about the system's reliability, or exhibit heterogeneous levels of sophistication. To this end, we explore a setting in which units follow recommendations imperfectly in Appendix G, and we find that our recommendation schemes are empirically robust to unit non-compliance.

## 3 The necessity of overlap

Suppose that the principal would like to estimate counterfactuals for some unit under intervention $d$. We begin by showing that the Unit Overlap Assumption must be satisfied for this unit under intervention $d$ in order for any algorithm to obtain consistent counterfactual estimates under the setting described in Section 2.

**Theorem 3.1.** *Fix a time horizon $T$, pre-intervention time period $T_0$, and number of donor units $n^{(0)}$. For any algorithm used to estimate the average post-intervention outcome under control for a test unit, there exists a problem instance such that the produced estimate has constant error whenever the Unit Overlap Assumption is not satisfied for the test unit under control, even as $T, T_0, n^{(0)} \to \infty$.*

The proof of Theorem 3.1 proceeds by constructing a family of problem instances under which the Unit Overlap Assumption is not satisfied, before showing that no algorithm can consistently estimate $\mathbb{E}[\bar{y}_{n^{(0)}+1,post}^{(0)}]$ on all instances in the family.

*Proof.* Consider the setting in which all type 0 units have latent factor $\mathbf{v}_0 = [0 \; 1]^{\top}$ and all type 1 units have latent factor $\mathbf{v}_1 = [1 \; 0]^{\top}$. Suppose that $\mathbf{u}_t^{(0)} = [1 \; 0]^{\top}$ if $t \mod 2 = 0$ and $\mathbf{u}_t^{(0)} = [0 \; 1]^{\top}$ if $t \mod 2 = 1$ if $t \leq T_0$. For all $t > T_0$, let $\mathbf{u}_t^{(0)} = [H \; 1]^{\top}$ for some $H$ in range $[-c, c]$, where $c > 0$. Furthermore, suppose that there is no noise in the unit outcomes. Under this setting, the (expected) outcomes for type 0 units under control are

$$\mathbb{E}[y_{0,t}^{(0)}] = \begin{cases} 0 & \text{if } t \mod 2 = 0, t \leq T_0 \\ 1 & \text{if } t \mod 2 = 1, t \leq T_0 \\ 1 & \text{if } t > T_0 \end{cases}$$

and the outcomes for type 1 units under control are

$$\mathbb{E}[y_{1,t}^{(0)}] = \begin{cases} 1 & \text{if } t \mod 2 = 0, t \leq T_0 \\ 0 & \text{if } t \mod 2 = 1, t \leq T_0 \\ H & \text{if } t > T_0. \end{cases}$$

Suppose that $H \sim \mathsf{Unif}[-c, c]$ and consider a principal who wants to estimate $\mathbb{E}[\bar{y}_{1,post}^{(0)}]$ using just the set of outcomes $\mathbb{E}[\mathbf{y}_{0,pre}]$, $\mathbb{E}[\mathbf{y}_{0,post}^{(0)}]$, and $\mathbb{E}[\mathbf{y}_{1,pre}]$. Since these outcomes do not contain any information about $H$, any estimator $\widehat{\mathbb{E}}[\bar{y}_{1,post}^{(0)}]$ cannot be a function of $H$ and thus will be at least a constant distance away from the true average post-intervention outcome $\mathbb{E}[\bar{y}_{1,post}^{(0)}]$ in expectation over $H$. That is,

$$\mathbb{E}_H \left| \mathbb{E}[\bar{y}_{1,post}^{(0)}] - \widehat{\mathbb{E}}[\bar{y}_{1,post}^{(0)}] \right| = \mathbb{E}_H \left| H - \widehat{\mathbb{E}}[\bar{y}_{1,post}^{(0)}] \right|$$
$$= \frac{1}{2c}(c^2 + (\widehat{\mathbb{E}}[\bar{y}_{1,post}^{(0)}])^2) \geq \frac{c}{2}.$$

Note that our choice of $T$ and $T_0$ was arbitrary, and that estimation does not improve as the number of donor units increases since (1) there is no noise in the outcomes and (2) all type 0 units have the same latent factor. Therefore any estimator for $\mathbb{E}[\bar{y}_{1,post}^{(0)}]$ is inconsistent in expectation over $H$ as $n^{(0)}, T, T_0 \to \infty$. This implies our desired result because if estimation error is constant in expectation over problem instances, there must exist at least one problem instance with constant estimation error. $\qquad \square$

### 3.1 On estimating unit-specific counterfactual outcomes

To infer the unit outcomes in the post-intervention period from outcomes in the pre-intervention period, we require the following *linear span inclusion* condition to hold on the latent factors associated with the pre- and post-intervention time-steps:

**Assumption 3.2** (Linear Span Inclusion). *For each intervention $d \in \{0, 1\}$ and time $t > T_0$, we assume that $\mathbf{u}_t^{(d)} \in \mathrm{span}\{\mathbf{u}_1^{(0)}, \mathbf{u}_2^{(0)}, \ldots, \mathbf{u}_{T_0}^{(0)}\}$.*

Much like the Unit Overlap Assumption, Assumption 3.2 is ubiquitous in the literature on robust synthetic control (see, e.g. Agarwal et al. (2020b; 2023); Harris et al. (2024)). In essence, Assumption 3.2 requires that the pre-intervention period be "sufficiently informative" about the post-intervention period. Indeed, Assumption 3.2 may be thought of as the "transpose" of the Unit Overlap Assumption, where the "overlap" condition is required to hold on the time-steps instead of on the units.[6] To gain intuition for why such an assumption is necessary, consider the limiting case in which $\mathbf{u}_t^{(0)} = \mathbf{0}_r$ for all $t \leq T_0$. Under such a setting, all expected unit outcomes in the pre-intervention time period will be 0, regardless of the underlying unit latent factors, which makes it impossible for the principal to infer anything non-trivial about a unit's post-intervention outcomes from looking at their pre-intervention outcomes alone.

**Vertical vs. horizontal regression.** The following proposition is an important structural result for our main algorithm, as it will allow us to transform the problem of estimating unit counterfactual outcomes from a regression over donor units (*vertical regression*) to a regression over pre-intervention outcomes (*horizontal regression*). While Proposition 3.3 is known in the literature on learning from panel data (e.g., (Agarwal et al., 2023; Harris et al., 2024)), we state it here for completeness.

**Proposition 3.3.** *Under Assumption 2.1 and 3.2, there exists a slope vector $\boldsymbol{\theta}^{(d)} \in \mathbb{R}^{T_0}$ such that the average expected post-intervention outcome of unit $i$ under intervention $d$ is given by $\mathbb{E}[\bar{y}_{i,post}^{(d)}] = \frac{1}{T_1}\langle\boldsymbol{\theta}^{(d)}, \mathbb{E}[y_{i,pre}]\rangle$. We assume that the principal has knowledge of a valid upper-bound $\Gamma$ on the $\ell_2$-norm of $\boldsymbol{\theta}^{(d)}$, i.e. $\|\boldsymbol{\theta}^{(d)}\|_2 \leq \Gamma$ for $d \in \{0, 1\}$.*

---

[6]Note that Assumption 3.2 is not a substitute for the Unit Overlap Assumption by Theorem 3.1.

---

**ALGORITHM 1:** Incentivizing Exploration for Synthetic Control: Type 1 units

---

**Input:** First stage length $N_0$, batch size $L$, number of batches $B$, failure probability $\delta$, gap $C \in (0,1)$

Provide no recommendation to first $N_0$ units.

**for** *batch* $b = 1, 2, \ldots, B$ **do**

    Select an explore index $i_b \in \llbracket L \rrbracket$ uniformly at random.

    **for** $j = 1, 2, \ldots, L$ **do**

        **if** $j = i_b$ **then**

            Recommend intervention $\widehat{d}_{N_0+(b-1)\cdot L+j} = 0$ to unit $N_0 + (b-1) \cdot L + j$.

        **else**

            **if** $\mu^{(0,1,l)} - \widehat{\mathbb{E}}[\bar{y}_{j,post}^{(1)}] \geq C$ **then**

                Recommend intervention $\widehat{d}_{N_0+(b-1)\cdot L+j} = 0$ to unit $N_0 + (b-1) \cdot L + j$.

            **else**

                Recommend intervention $\widehat{d}_{N_0+(b-1)\cdot L+j} = 1$ to unit $N_0 + (b-1) \cdot L + j$.

---

Given historical data of the form $\{\mathbf{y}_{j,pre}, \mathbf{y}_{j,post}^{(d_j)}\}_{j=1}^{i-1}$, we can estimate $\boldsymbol{\theta}^{(d)}$ as $\widehat{\boldsymbol{\theta}}_i^{(d)}$ using some estimation procedure, which in turn allows us to estimate $\mathbb{E}[\bar{y}_{i,post}^{(d)}]$ as $\widehat{\mathbb{E}}[\bar{y}_{i,post}^{(d)}] := \frac{1}{T_1}\langle\widehat{\boldsymbol{\theta}}_i^{(d)}, y_{i,pre}\rangle$ for $d \in \{0,1\}$. While this is not a synthetic control method in the traditional sense, as it does not construct a "synthetic control" trajectory from the trajectories of the donor units, it turns out that the point estimate it produces (and assumptions it requires) are equivalent to that of a vertical regression-based SCM in many settings (including ours). See Shen et al. (2022) for more background on the similarities and differences between using horizontal regression-based approaches (like ours) and vertical regression-based approaches (like traditional SCMs) in panel data settings. We choose to use a horizontal regression-based approach because it is easier to work with in our setting, where units arrive sequentially (recall Figure 1). Importantly, we would still need the Unit Overlap Assumption to be satisfied for intervention $d$ if we used a vertical regression-based method.

For the reader familiar with bandit algorithms, it may be useful to draw the following connection between our setting and that of linear contextual bandits. In particular, it is sometimes helpful to think of $\mathbb{E}[y_{i,pre}]$ as the "context" of unit $i$, and $\mathbb{E}[\bar{y}_{i,post}^{(d)}]$ as the principal's expected reward of assigning intervention $d$ given $\mathbb{E}[y_{i,pre}]$. Since we observe $y_{i,pre}$ instead of $\mathbb{E}[y_{i,pre}]$ we cannot apply linear contextual bandit algorithms to our panel data setting out-of-the-box. However as we will show in Section 4, one can combine ideas from incentivizing exploration in contextual bandits with principled ways of handling the noise in the pre-intervention outcomes to incentivize exploration in panel data settings.

## 4 Incentivized exploration for synthetic control

In this section, we turn our focus to incentivizing type 1 units (i.e. units who *a priori* prefer the treatment) to remain under control in the post-intervention period. The methods we present may also be applied to incentivize type 0 units (i.e. units who *a priori* prefer the control) to take the treatment. We focus on incentivizing control amongst type 1 units in order to be in line with the literature on synthetic control, which aims to estimate counterfactual unit outcomes under control.

The goal of the principal is to design a sequence of recommendation policies that convinces enough units of type 1 to select the control in the post-treatment period, such that the Unit Overlap Assumption is satisfied for both interventions for all units of type 1. At a high level, the principal is able to incentivize units to explore because they have more information than any one particular unit (as they view the realized history $H_i$ before making a recommendation), and so they can selectively reveal information in order to persuade units to take interventions which they would not normally take. Our algorithm (Algorithm 1) is inspired by the "detail free" algorithm for incentivizing exploration in multi-armed bandits in Mansour et al. (2020). Furthermore, through our discussion in Section 3, one may view Algorithm 1 as an algorithm for incentivizing exploration in linear contextual bandit settings with measurement error in the context.

The recommendation policy in Algorithm 1 is split into two stages. In the first stage, the principal provides no recommendations to the first $N_0$ units. Left to their own devices, these units take their preferred intervention according to their prior belief: type 0 units take control, and type 1 units take the treatment. We choose $N_0$ such that it is large enough for the Unit Overlap Assumption to be satisfied for all units of type 1 under the treatment with high probability.[7]

In the second stage, we use the initial data collected during the first stage to construct an estimator of the average expected post-intervention outcome for type 1 units under treatment using PCR, as described in Section 3.1. By dividing the total number of units in the second stage into phases of $L$ rounds each, the principal can randomly "hide" one *explore* recommendation amongst $L-1$ *exploit* recommendations. When the principal sends an exploit recommendation to unit $i$, they recommend the intervention which would result in the highest estimated average post-intervention outcome for unit $i$, where unit $i$'s post-intervention outcomes under treatment are estimated using the data collected during the first stage. On the other hand, the principal recommends that unit $i$ take the control whenever they send an explore recommendation.

Thus under Algorithm 1, if a type 1 unit receives a recommendation to take the treatment, they will always follow the recommendation since they can infer they must have received an exploit recommendation. However if a type 1 unit receives a recommendation to take the control, they will be unsure if they have received an explore or an exploit recommendation, and can therefore be incentivized to follow the recommendation as long as $L$ is set to be large enough (i.e. the probability of an explore recommendation is sufficiently low). This works because the sender's estimate of $\mathbb{E}[\bar{y}_{i,post}^{(1)}]$ is more informed than unit $i$'s prior belief, due to the history of outcomes $H_i$ that the principal has observed.

Our algorithm requires that bounds on various *aggregate* statistics about the underlying unit population are common knowledge amongst the principal and all units. Such assumptions are standard in the literature on incentivized exploration.

We are now ready to present our main result: a way to initialize Algorithm 1 to ensure that the Unit Overlap Assumption is satisfied with high probability.

**Theorem 4.1** (Informal; detailed version in Theorem C.5). *Under Assumption C.3, if the number of initial units $N_0$ and the batch size $L$ are chosen to be sufficiently large, then Algorithm 1 satisfies the BIC property (Definition 2.7).*

*Moreover, if the number of batches $B$ is chosen to be sufficiently large, then the conditions of the Unit Overlap Assumption will be satisfied simultaneously for all type 1 units under control w.h.p.*

*Summary of Assumption C.3*: Assumption C.3 posits that the principal and units know valid upper- and lower-bounds on various population-level statistics about the underlying unit population (e.g., the fraction of units of each type, the prior mean for each type and intervention). As these are population-level statistics, in practice, the principal may be able to learn these bounds from samples of the data and broadcast them to the unit population. See Appendix C.2 for more details.

*Proof Sketch.* See Appendix C for complete proof details. At a high level, the analysis follows by expressing the compliance condition for type 1 units as different cases depending on the principal's recommendation. In particular, a type 1 unit could receive recommendation $\hat{d}_i = 0$ for two reasons: (1) Under event $\xi_{C,i}$ when control is indeed the better intervention according to the prior and the observed outcomes of previous units, or (2) when the unit is randomly selected as an explore unit. Using bounds on the probabilities of these two events occurring, we can derive a condition on the minimum phase length $L$ such that a unit's expected gain from exploiting (when the event $\xi_{C,i}$ happens) exceeds the expected (unit) loss from exploring. We then further simplify the condition on the phase length $L$ so that it is computable by the principal by leveraging existing finite sample guarantees for PCR using the samples collected in the first stage when no recommendations are given. $\qquad\square$

---

[7]The Unit Overlap Assumption will always be satisfied w.h.p. for type 1 units under the treatment after seeing sufficiently-many type 1 units. The hard part is convincing enough type 1 units to remain under control so that the Unit Overlap Assumption is also satisfied for control.

Theorem 4.1 says that after running Algorithm 1 with optimally-chosen parameters, the Unit Overlap Assumption will be satisfied for all type 1 units *under both interventions* with high probability. Therefore after running Algorithm 1 on sufficiently-many units, the principal can use off-the-shelf SCMs (e.g. Agarwal et al. (2020b; 2023)) to obtain counterfactual estimates with valid confidence intervals for all type 1 units under control (w.h.p.). Note that all parameters required to run Algorithm 1 are computable by the principal in polynomial time. See Example C.6 for a concrete instantiation of Algorithm 1.

## 5 Testing whether the Unit Overlap Assumption holds

So far, our focus has been on designing algorithms to incentivize subpopulations of units to take interventions for which the Unit Overlap Assumption does not initially hold. In this section we study the complementary problem of designing a procedure for determining whether the Unit Overlap Assumption holds for a given test unit and set of donor units. In particular, we provide two simple hypothesis tests for deciding whether the Unit Overlap Assumption holds, by leveraging existing finite sample guarantees for PCR. Our first hypothesis test is non-asymptotic, and thus is valid under any amount of data. While our second hypothesis test is asymptotic and therefore valid only in the limit, we anticipate that it may be of more practical use.

### 5.1 A non-asymptotic hypothesis test

Our first hypothesis test proceeds as follows: Using the PCR techniques described in Appendix B, we learn a linear relationship between the first $T_0/2$ time-steps in the pre-intervention time period and the average outcome in the second half of the pre-intervention time period using data from the donor units. Using this learned relationship, we then compare our estimate for the test unit with their *true* average outcome in the second half of the pre-intervention time period. If the difference between the two is larger than the confidence interval given by PCR (plus an additional term to account for the noise), we can conclude that the Unit Overlap Assumption is not satisfied with high probability.

In order for our hypothesis test to be valid we require the following assumption, which is analogous to Assumption 3.2.

**Assumption 5.1** (Linear Span Inclusion, revisited). *For every time-step $t$ such that $T_0/2 < t \leq T_0$, we assume that $\mathbf{u}_t^{(0)} \in \mathrm{span}\{\mathbf{u}_1^{(0)}, \ldots, \mathbf{u}_{T_0/2}^{(0)}\}$.*

While Assumption 3.2 requires that the post-intervention latent factors for any intervention fall within the linear span of the pre-intervention latent factors, Assumption 5.1 requires that the second half of the pre-intervention latent factors fall within the linear span of the first half. This assumption allows us to obtain valid confidence bounds when regressing the second half of the pre-intervention outcomes on the first half. Let $y_{i,pre'} := [y_{i,1}^{(0)}, \ldots, y_{i,T_0/2}^{(0)}]$, $y_{i,pre''} := [y_{i,T_0/2+1}^{(0)}, \ldots, y_{i,T_0}^{(0)}]$, and $\bar{y}_{i,pre''} := \frac{2}{T_0}\sum_{t=T_0/2+1}^{T_0} y_{i,t}^{(0)}$. Under Assumption 5.1 we can estimate $\mathbb{E}[\bar{y}_{i,pre''}]$ as $\widehat{\mathbb{E}}[\bar{y}_{i,pre''}] := \langle \widehat{\boldsymbol{\theta}}_i, y_{i,pre'} \rangle$, where $\widehat{\boldsymbol{\theta}}_i$ is computed using the PCR techniques in Appendix B.

We are now ready to introduce our hypothesis test. In what follows, we will leverage the high-probability confidence interval of Agarwal et al. (2023) (Theorem C.1). Under Assumption 5.1, we can apply Theorem C.1 out-of-the-box to estimate $\mathbb{E}[\bar{y}_{i,pre''}]$ using the first $T_0/2$ time-steps as the pre-intervention period and the next $T_0/2$ time-steps as the post-intervention period.

**Hypothesis test.** Consider the hypothesis that Unit Overlap Assumption is not satisfied for test unit $n$.

1. Using donor units $\mathcal{I}$ and test unit $n$, compute $\widehat{\mathbb{E}}[\bar{y}_{n,pre''}]$ via the PCR method of Appendix B.

2. Let $\alpha(\delta)$ be the confidence bound given in Theorem C.1 s.t. $|\mathbb{E}[\bar{y}_{i,pre''}] - \widehat{\mathbb{E}}[\bar{y}_{n,pre''}]| \leq \alpha(\delta)$ with probability at least $1 - \delta$. Specifically, compute $\alpha(\delta) + 2\sigma\sqrt{\frac{\log(1/\delta)}{T_0}}$ for confidence level $\delta$ by applying Theorem C.1 with the first $T_0/2$ time-steps as the pre-intervention period and the next $T_0/2$ time-steps as the post-intervention period.

3. If $|\widehat{\mathbb{E}}[\bar{y}_{n,pre''}] - \bar{y}_{n,pre''}| > \alpha(\delta) + 2\sigma\sqrt{\frac{\log(1/\delta)}{T_0}}$ then accept the hypothesis. Otherwise, reject.

**Theorem 5.2.** *Under Assumption 5.1, if the Unit Overlap Assumption is satisfied for unit $n$, then* $|\widehat{\mathbb{E}}[\bar{y}_{n,pre''}] - \bar{y}_{n,pre''}| \leq \alpha(\delta) + 2\sigma\sqrt{\frac{\log(1/\delta)}{T_0}}$ *with probability $1 - \mathcal{O}(\delta)$, where $\alpha(\delta)$ is the high-probability confidence interval which is defined in Theorem C.1 when using the first $T_0/2$ time-steps as the pre-intervention period and the next $T_0/2$ time-steps as the post-intervention period.*

Therefore we can conclude that if $|\widehat{\mathbb{E}}[\bar{y}_{i,pre''}] - \bar{y}_{i,pre''}| > \alpha(\delta) + 2\sigma\sqrt{\frac{\log(1/\delta)}{T_0}}$, then the Unit Overlap Assumption will not be satisfied for unit $i$ w.h.p.

### 5.2 An asymptotic hypothesis test

Next we present a hypothesis test which, while only valid in the limit, may be of more practical use when compared to the non-asymptotic hypothesis test of Section 5.1. At a high level, our asymptotic hypothesis test leverages Assumption 5.1 and the guarantees for synthetic interventions (SI) in Agarwal et al. (2020b) to determine whether the Unit Overlap Assumption holds for a given test unit $n$. In particular, their results imply that under Assumption 5.1 and the Unit Overlap Assumption, a rescaled version of $\widehat{\mathbb{E}}[\bar{y}_{n,pre''}] - \mathbb{E}[\bar{y}_{n,pre''}]$ converges in distribution to the standard normal distribution as $|\mathcal{I}|, T_0, T_1 \to \infty$. While the rescaling amount depends on quantities which are not computable, a relatively simple application of the continuous mapping theorem and Slutsky's theorem imply that we can compute a valid test statistic which converges in distribution to the standard normal.

**Notation.** In what follows, let $\omega^{(n,0)} \in \mathbb{R}^{|\mathcal{I}|}$ be the linear relationship between the test unit $n$ and donor units $\mathcal{I}$ which is known to exist under the Unit Overlap Assumption, i.e. $\mathbb{E}[y_{n,t}^{(0)}] = \sum_{i \in \mathcal{I}} \omega^{(n,0)}[i] \cdot \mathbb{E}[y_{i,t}^{(0)}]$. Let $\widetilde{\omega}^{(n,0)}$ be the projection of $\omega^{(n,0)}$ onto the subspace spanned by the latent factors of the $\mathcal{I}$ test units. Let $\widehat{\omega}^{(n,0)}$ be the estimate of $\widetilde{\omega}^{(n,0)}$ and $\widehat{\sigma}$ be the estimate of the standard deviation of the noise terms $\{\epsilon_{i,t} : i \in \mathcal{I}, t \in [\![T]\!]\}$ given by SC. Finally, let $z_\alpha$ be the z score at significance level $\alpha \in (0,1)$, i.e. $\alpha = \mathbb{P}(x \leq z_\alpha)$, where $x \sim \mathcal{N}(0,1)$.

**Hypothesis test.** Consider the hypothesis that the Unit Overlap Assumption is not satisfied for a test unit $n$.

1. Using donor units $\mathcal{I}$ and test unit $n$, compute $\widehat{\mathbb{E}}[\bar{y}_{n,pre''}]$ and $\widehat{\omega}^{(n,0)}$ using SI.
2. Accept the hypothesis only if $\frac{\sqrt{T_1}}{\widehat{\sigma}\|\widehat{\omega}^{(n,0)}\|_2}|\widehat{\mathbb{E}}[\bar{y}_{n,pre''}] - \bar{y}_{n,pre''}| > z_{0.95}$.

**Theorem 5.3** (Informal; detailed version in Theorem F.1). *Under Assumption 5.1, if $\|\widetilde{\omega}^{(n,0)}\|_2$ is sufficiently large and $\widehat{\omega}^{(n,0)} \to \widetilde{\omega}^{(n,0)}$ at a sufficiently fast rate, then as $|\mathcal{I}|, T_0, T_1 \to \infty$ the Asymptotic Hypothesis Test falsely accepts the hypothesis with probability at most 5%.*

## 6 Numerical simulations

In this section, we complement our theoretical results with a numerical comparison to standard SCMs which do not take incentives into consideration.

**Experiment Setup.** We consider the setting of Section 4: There are two interventions and two unit types. Type 1 units prefer the treatment and type 0 units prefer the control. Our goal is to incentivize type 1 units to take the control so we can obtain accurate counterfactual estimates of all units under control. See Appendix G for details about our data-generating process.

We apply the asymptotic hypothesis test of Section 5.2 using all type 0 units as the donor units and a single type 1 unit as the test unit. We accept the null hypothesis, as the test statistic $\frac{\sqrt{T_1}}{\widehat{\sigma}\|\widehat{\omega}^{(n,0)}\|_2}|\widehat{\mathbb{E}}[\bar{y}_{n,pre''}] - \bar{y}_{n,pre''}| \approx$ 2.24, which is much larger than the value of $z_{0.95} \approx 1.96$ needed for acceptance. That is, the asymptotic test (correctly) returns that the Unit Overlap Assumption is not satisfied for a test unit of type 1 under control.

Using Theorem C.5 (the formal version of Theorem 4.1), we can calculate a lower bound on the phase length $L$ of Algorithm 1 such that the BIC condition (Definition 2.7) is satisfied for units of type 1 who

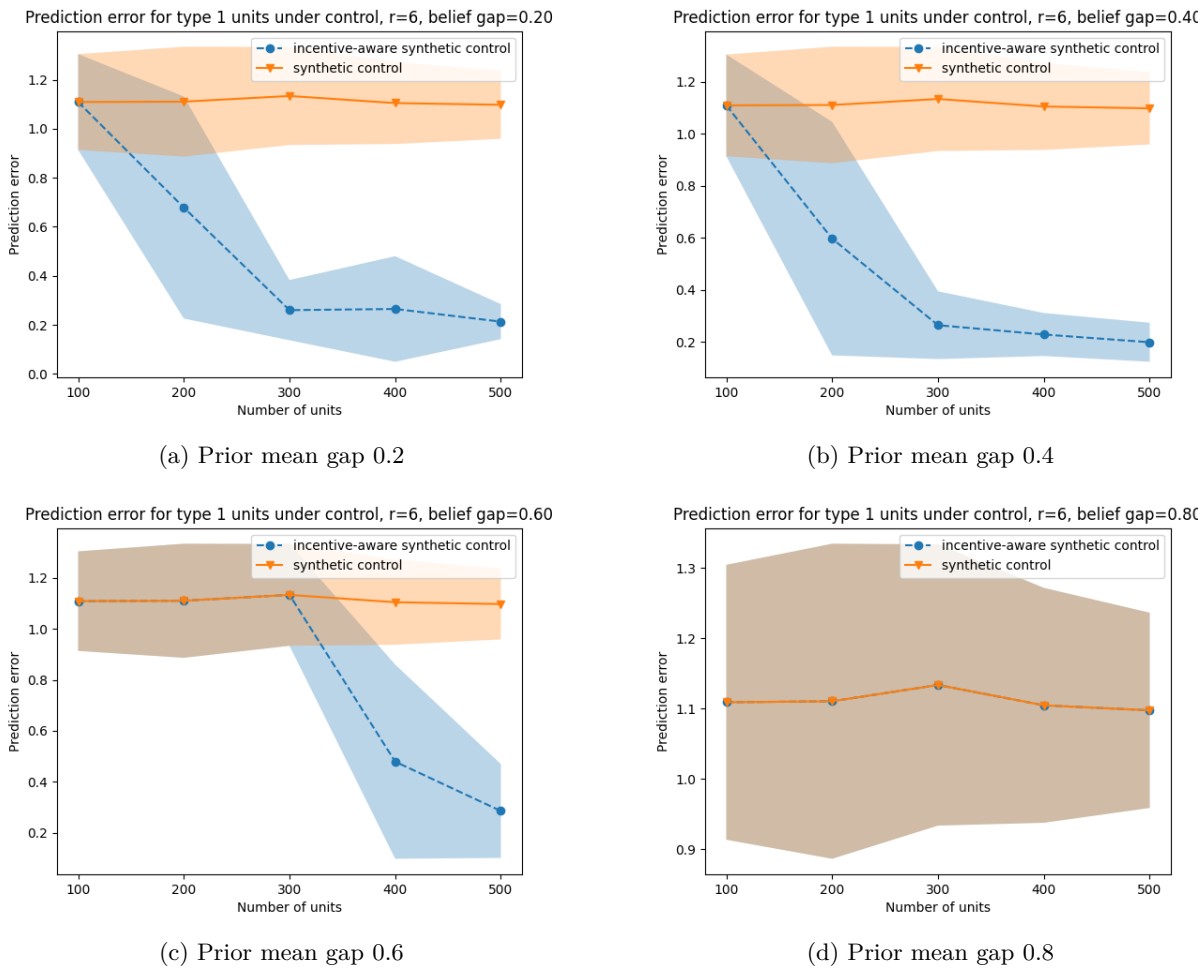

(a) Prior mean gap 0.2

(b) Prior mean gap 0.4

(c) Prior mean gap 0.6

(d) Prior mean gap 0.8

Figure 2: Counterfactual estimation error for units of type 1 under control using Algorithm 1 (blue) and synthetic control without incentives (orange) with different gaps between the prior mean reward of control and treatment. Results are averaged over 50 runs, with the shaded regions representing one standard deviation.

are recommended the control. We run three different sets of simulations and compare the estimation error $|\widehat{\mathbb{E}}\bar{y}_{n,post}^{(0)} - \mathbb{E}\bar{y}_{n,post}^{(0)}|$ for a new type 1 unit $n$ under control when incentivizing exploration using Algorithm 1 (blue) to the estimation error without incentivizing exploration (orange). For both our method and the incentive-unaware ablation, we use the adaptive PCR method of Agarwal et al. (2023) to estimate counterfactuals. All experiments are repeated 50 times and we report both the average prediction error and the standard deviation for estimating the post-treatment outcome of type 1 units under control.

**Varying Strength of Belief.** In Figure 2, we explore the effects of changing the units' "strength of beliefs". Specifically, we vary the gap between the prior mean outcome under control and treatment for type 1 units by setting $\mu_i^{(1)} - \mu_i^{(0)} \in \{0.2, 0.4, 0.6, 0.8\}$. As the gap decreases it becomes easier to persuade units to explore, and therefore our counterfactual estimates improve.

**Varying Latent Factor Size.** In Figure 3, we vary $r \in \{2, 4, 6, 8\}$ while keeping all other parameters of the problem instance constant. Increasing $r$ increases the batch size $L$, so we see a modest decrease in performance as $r$ increases.

### 6.1 Hypothesis test

In order to test the validity of the asymptotic hypothesis test in Section 5.2, we use synthetic data for which we know the underlying data-generating process.

Specifically, we create two types of synthetic datasets: one where the Unit Overlap Assumption holds ("UOA data"), and one where it does not ("non-UOA data"). For simplicity, we consider a setting where there are two interventions and two unit types.

For non-UOA data, we use the same data-generating process as described above, where if unit $i$ is of type 1 (resp. type 0), we generate a latent vector $\mathbf{v}_i = [0 \ \cdots \ 0 \ v_i[1] \ \cdots \ v_i[r/2]]$ (resp. $\mathbf{v}_i = [v_i[1] \ \cdots \ v_i[r/2] \ 0 \ \cdots \ 0])$, where $\forall j \in [r/2] : v_i[j] \sim \mathsf{Unif}(0,1)$. This dataset then contains 500 units of each type. On the other hand, UOA data contains 1000 Type 0 units only, where the latent vector is generated the same way as Type 0 units in non-UOA data. The pre-treatment and post-treatment latent factors are generated in the same manner as above.

Using this data-generating process, we generate 500 datasets of each type (for a total of 1000 datasets), and run the asymptotic hypothesis test in Section 5.2 for each dataset. If we accept the null hypothesis, i.e., the Unit Overlap Assumption is predicted not to be satisfied according to our test, then we say the 'predicted' label of the dataset is 1; otherwise, the 'predicted' label is 0. We say that a dataset has (true) label 1 if the Unit Overlap Assumption is not satisfied, and label 0 if it is.

We report the true positive rate (TPR) and false positive rate (FPR) for this experiment. The resulting confusion matrix is:

$$\begin{bmatrix} 475 & 25 \\ 22 & 478 \end{bmatrix}$$

with the $TPR = 0.956$ and $FPR = 0.05$. This experimental result supports Theorem 5.3, where the FPR is at most 0.05.

## 7 Conclusion

We study the problem of non-compliance when estimating counterfactual outcomes using panel data. Our focus is on synthetic control methods, which canonically require an overlap assumption on the unit outcomes in order to provide valid finite sample guarantees. We shed light on this often overlooked assumption, and provide an algorithm for incentivizing units to explore different interventions using tools from information design and online learning. After running our algorithm, the Unit Overlap Assumption will be satisfied with high probability, which allows for the principal to obtain valid finite sample guarantees for all units when using off-the-shelf SCMs to estimate counterfactual outcomes under control. We also extend our algorithm to satisfy the Unit Overlap Assumption for all units and all interventions when there are more than two interventions, and we provide two hypothesis tests for determining if the Unit Overlap Assumption hold for a given test unit and set of donor units. Finally, we complement our theoretical findings with numerical simulations, and observe that our procedure for estimating counterfactual outcomes produces significantly more accurate estimates when compared to existing methods which do not take unit incentives into consideration.

There are several exciting directions for future research. In some multi-armed bandit settings, lower bounds are known on the sample complexity of incentivizing exploration (Sellke & Slivkins (2023); Sellke (2023)). It would be interesting to prove analogous bounds on the number of units required to incentivize sufficient exploration for the Unit Overlap Assumption to be satisfied for all units and all interventions in our setting. Another avenue for future research is to design difference-in-differences or clustering-based algorithms for incentivizing exploration in other causal inference settings. Finally it would be interesting to relax other assumptions typically needed for synthetic control, such as the linear span inclusion assumption on the time-step latent factors.

## Acknowledgments

We would like to thank the anonymous reviewers for their helpful feedback. For part of this work, DN was a Ph.D. student at the University of Minnesota and KH was a Ph.D. student at Carnegie Mellon University. VS was supported by NSF Award IIS-2337916 and ZSW was supported by an NSF CAREER Award.

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

## A   Additional related work

There has been recent interest in causal inference on characterizing optimal treatment policies which must satisfy some additional constraint(s). For example, Luedtke & van der Laan (2016) study a setting in which the treatment supply is limited. Much like us, Qiu et al. (2021) consider a setting in which intervening on the treatment is not possible, but encouraging treatment is feasible. However their focus is on the setting in which the treatment is a limited resource, so their goal is to encourage treatment to those who would benefit from it the most. Since they do not consider panel data and place different behavioral assumptions on the individuals under intervention, the tools and techniques we use to persuade units in our setting differ significantly from theirs. Finally, Qiu et al. (2022); Sun et al. (2021) consider settings where there is some uncertain *cost* associated with treatment.

Within the literature on IE, the work most related to ours on a conceptual level is that of Li & Slivkins (2022), who consider the problem of incentivizing exploration in clinical trial settings. Like us, Li & Slivkins (2022) study a setting in which the principal would like to incentivize agents to explore different treatments. However, their goal is to estimate *population-level* statistics about each intervention, while we are interested in estimating *unit-specific* counterfactuals under different interventions using panel data. As a result, while our high-level motivations are somewhat similar, the tools and techniques we use to obtain our results differ significantly. On a technical level, our mechanisms are somewhat similar to the initial exploration phase in Mansour et al. (2020), although we consider a more general setting where unit outcomes may vary over time, in contrast to the simpler multi-armed bandit setting they consider.

Finally, our work is conceptually related to the literature on *causal bandits* (Lattimore et al., 2016), which is a sequential decision-making setting where actions correspond to interventions on a causal graph. As we mention in Section 3, our setting may be viewed as a linear contextual bandit setting, where interventions are actions and the principal observes a noisy version of the context. Within the literature on causal bandits, papers on linear causal bandits (where the underlying causal system is assumed to be linear) are perhaps the most similar to our work (e.g. Yan & Tajer (2024); Yan et al. (2024); Varici et al. (2023)). However unlike this line of work, we do not study regret minimization under a causal graph, but instead focus on the problem of identification in the panel data setting. Moreover, our focus is on the regime in which compliance cannot be enforced and units are free to self-select their interventions.

## B   Background on principal component regression

The specific algorithm we use to estimate $\boldsymbol{\theta}^{(d)}$ is called *principal component regression* (PCR). At a high level, PCR "de-noises" the matrix of covariates (in our case, the set of all pre-treatment outcomes for all donor units seen so far who underwent intervention $d$) using hard singular value thresholding, before regressing the (now-denoised) covariates on the outcomes of interest (in our case, the average post-intervention outcome under intervention $d$).

Let $\mathcal{I}_i^{(d)}$ be the set of units who have received intervention $d$ before unit $i$ arrives, and let $n_i^{(d)}$ be the number of such units. We use

$$Y_{pre,i}^{(d)} := [\mathbf{y}_{j,pre}^\top \; : \; j \in \mathcal{I}_i^{(d)}] \in \mathbb{R}^{n_i^{(d)} \times T_0}$$

to denote the matrix of pre-treatment outcomes corresponding to the subset of units who have undergone intervention $d$ before unit $i$ arrives, and

$$Y_{post,i}^{(d)} := \left[ \sum_{t=T_0+1}^{T} y_{j,t}^{(d)} : j \in \mathcal{I}_i^{(d)} \right] \in \mathbb{R}^{n_i^{(d)} \times 1}$$

be the vector of the sum of post-intervention outcomes for the subset of units who have undergone intervention $d$ before unit $i$ arrives. We denote the singular value decomposition of $Y_{pre,i}^{(d)}$ as

$$Y_{pre,i}^{(d)} = \sum_{\ell=1}^{n_i^{(d)} \wedge T_0} s_\ell^{(d)} \widehat{\mathbf{u}}_\ell^{(d)} (\widehat{\mathbf{v}}_\ell^{(d)})^\top,$$

where $\{s_\ell^{(d)}\}_{\ell=1}^{n_i^{(d)} \wedge T_0}$ are the singular values of $Y_{pre,i}^{(d)}$, and $\widehat{\mathbf{u}}_\ell^{(d)}$ and $\widehat{\mathbf{v}}_\ell^{(d)}$ are orthonormal column vectors. We assume that the singular values are ordered such that $s_1(Y_{pre,i}^{(d)}) \geq \cdots \geq s_{n_i^{(d)} \wedge T_0}(Y_{pre,i}^{(d)}) \geq 0$. For some threshold value $r$, we use

$$\widehat{Y}_{pre,i}^{(d)} := \sum_{\ell=1}^{r} s_\ell^{(d)} \widehat{\mathbf{u}}_\ell^{(d)} (\widehat{\mathbf{v}}_\ell^{(d)})^\top$$

to refer to the truncation of $Y_{pre,i}^{(d)}$ to its top $r$ singular values. We define the projection matrix onto the subspace spanned by the top $r$ right singular vectors as $\widehat{\mathbf{P}}_{i,r}^{(d)} \in \mathbb{R}^{r \times r}$, given by $\widehat{\mathbf{P}}_{i,r}^{(d)} := \sum_{\ell=1}^{r} \widehat{\mathbf{v}}_\ell^{(d)} \left( \widehat{\mathbf{v}}_\ell^{(d)} \right)^\top$. Equipped with this notation, we are now ready to define the procedure for estimating $\boldsymbol{\theta}^{(d)}$ using (regularized) principal component regression.

**Definition B.1** (Regularized Principal Component Regression). *Given regularization parameter $\rho \geq 0$ and truncation level $r \in \mathbb{N}$, for $d \in \{0,1\}$ and $i \geq 1$, let $\mathcal{V}_i^{(d)} := \left( \widehat{Y}_{pre,i}^{(d)} \right)^\top \widehat{Y}_{pre,i}^{(d)} + \rho \widehat{\mathbf{P}}_{i,r}^{(d)}$. Then, regularized PCR estimates $\boldsymbol{\theta}^{(d)}$ as*

$$\widehat{\boldsymbol{\theta}}_i^{(d)} := \left( \mathcal{V}_i^{(d)} \right)^{-1} \widehat{Y}_{pre,i}^{(d)} Y_{i,post}^{(d)},$$

*where $\boldsymbol{\theta}^{(d)}$ is defined as in Proposition 3.3. The average post-intervention outcome for unit $i$ under intervention $d$ may then be estimated as $\widehat{\mathbb{E}}[\bar{y}_{i,post}^{(d)}] := \frac{1}{T_1} \langle \widehat{\boldsymbol{\theta}}_i^{(d)}, y_{i,pre} \rangle$.*

Observe that if $\rho = 0$, then $\mathcal{V}_i^{(d)} := \left( \widehat{Y}_{pre,i}^{(d)} \right)^\top \widehat{Y}_{pre,i}^{(d)}$ and we recover the non-regularized version of PCR. Under the Unit Overlap Assumption and Assumption 3.2, work on robust synthetic control (see Section 1.1) uses (regularized) PCR to obtain consistent estimates for counterfactual post-intervention outcomes under different interventions. Intuitively, the Unit Overlap Assumption is required for consistent estimation because there needs to be sufficient information about the test unit contained in the data from donor units in order to accurately make predictions about the test unit using a model learned from the donor units' data.

## C    Appendix for Section 4: Incentivized exploration for synthetic control

### C.1    Causal parameter recovery

**Theorem C.1** (Theorem G.3 of Agarwal et al. (2023)). *Let $\delta \in (0,1)$ be an arbitrary confidence parameter and $\rho > 0$ be chosen to be sufficiently small. Further, assume that Assumption 2.1 and Unit Overlap Assumption are satisfied, there is some $i_0 \geq 1$ such that $\mathrm{rank}(\mathbf{X}_{i_0}](d)) = r$, and $\mathrm{snr}_i(d) \geq 2$ for all $i \geq i_0$.*

*Then, with probability at least $1 - \mathcal{O}(k\delta)$, simultaneously for all interventions $d \in [k]_0$,*

$$\left| \widehat{\mathbb{E}}[\bar{y}_{i,post}^{(d)}] - \bar{\mathbb{E}}[y_{i,post}^{(d)}] \right| \leq \alpha(\delta),$$

$$\text{where } \alpha(\delta) := \frac{3\sqrt{T_0}}{\widehat{\text{snr}}_i(d)} \left( \frac{L(\sqrt{74} + 12\sqrt{6}\kappa(\mathbf{Z}_i(d)))}{(T - T_0) \cdot \widehat{\text{snr}}_i(d)} + \frac{\sqrt{2\text{err}_i(d)}}{\sqrt{T - T_0} \cdot \sigma_r(\mathbf{Z}_i(d))} \right)$$

$$+ \frac{2L\sqrt{24T_0}}{(T - T_0) \cdot \widehat{\text{snr}}_i(d)} + \frac{12L\kappa(\mathbf{Z}_i(d))\sqrt{3T_0}}{(T - T_0) \cdot \widehat{\text{snr}}_i(d)} + \frac{2\sqrt{\text{err}_i(d)}}{\sqrt{T - T_0} \cdot \sigma_r(\mathbf{Z}_i(d))}$$

$$+ \frac{L\sigma\sqrt{2\log(k/\delta)}}{T - T_0} + \frac{L\sigma\sqrt{148\log(k/\delta)}}{\widehat{\text{snr}}_i(d)(T - T_0)} + \frac{24\sigma\kappa(\mathbf{Z}_i(d))\sqrt{6\log(k/\delta)}}{\widehat{\text{snr}}_i(d)(T - T_0)}$$

$$+ \frac{2\sigma\sqrt{\text{err}_i(d)\log(k/\delta)}}{\sigma_r(\mathbf{Z}_i(d))\sqrt{T - T_0}}$$

*where $\widehat{\mathbb{E}}[\bar{y}_{i,post}^{(d)}] := \frac{1}{T - T_0} \cdot \langle \widehat{\theta}_i(d), y_{i,pre} \rangle$ is the estimated average post-intervention outcome for unit $i$ under intervention $d$ and $\left\| \theta_i^{(d)} \right\| \leq L$.*

**Corollary C.2.** *Given a gap $\epsilon$ and the same assumptions as in Theorem C.1, the probability that $\left| \widehat{\mathbb{E}}[\bar{y}_{i,post}^{(d)}] - \mathbb{E}[\bar{y}_{i,post}^{(d)}] \right| \leq \epsilon$ is at least $1 - \delta$, where*

$$\delta \leq \frac{\log(n^{(d)}) \vee k}{\exp\left( \left( \sqrt{\frac{\sigma_{r_d}(\mathbf{Z}_i(d))\epsilon - (A + F)(\sqrt{n^{(d)}} + \sqrt{T_0})}{D} + \frac{\alpha^2}{4D^2}} - \frac{\alpha}{2D} \right)^2 \right)}$$

*with*

- $A = 3\sqrt{T_0} \left( \frac{\left\| \theta_i^{(d)} \right\|(\sqrt{74} + 12\sqrt{6}\kappa(\mathbf{Z}_i(d)))}{T - T_0} + \frac{\sqrt{2}}{\sqrt{T - T_0}} \right),$

- $F = \frac{2\left\| \theta_i^{(d)} \right\|\sqrt{24T_0}}{T - T_0} + \frac{12\left\| \theta_i^{(d)} \right\|\kappa(\mathbf{Z}_i(d))\sqrt{3T_0}}{T - T_0} + \frac{2}{\sqrt{T - T_0}},$

- $D = \frac{\left\| \theta_i^{(d)} \right\|\sigma\sqrt{148}}{T - T_0} + \frac{24\sigma\kappa(\mathbf{Z}_i(d))\sqrt{6}}{T - T_0} + \frac{2\sigma}{\sqrt{T - T_0}},$

- $E = \frac{\sqrt{2}\left\| \theta_i^{(d)} \right\|\sigma}{T - T_0},$

- $\alpha = A + F + D(\sqrt{n^{(d)}} + \sqrt{T_0}) + \sigma_{r_d}(\mathbf{Z}_i(d))E.$

*Proof.* We begin by setting the right-hand side of Theorem C.1 to be $\epsilon$. The goal is to write the failure probability $\delta$ as a function of $\epsilon$. Then, using the notations above, we can write

$$\epsilon = \frac{A}{(\widehat{\text{snr}}_i(d))^2} + \frac{F}{\widehat{\text{snr}}_i(d)} + \frac{D\sqrt{\log(k/\delta)}}{\widehat{\text{snr}}_i(d)} + E\sqrt{\log(k/\delta)}$$

First, we take a look at the signal-to-noise ratio $\widehat{\text{snr}}_i(d)$. By definition, we have:

$$\widehat{\text{snr}}_i(d) = \frac{\sigma_r(\mathbf{Z}_i(d))}{U_i}$$

$$= \frac{\sigma_r(\mathbf{Z}_i(d))}{\sqrt{n^{(d)}} + \sqrt{T_0} + \sqrt{\log(\log(n^{(d)})/\delta)}}$$

Hence,

$$\frac{1}{\widehat{\mathrm{snr}}_i(d)} = \frac{\sqrt{n^{(d)}} + \sqrt{T_0} + \sqrt{\log(\log(n^{(d)})/\delta)}}{\sigma_r(\mathbf{Z}_i(d))}$$

Observe that since $\widehat{\mathrm{snr}}_i(d) \geq 2$, we have $\frac{1}{(\widehat{\mathrm{snr}}_i(d))^2} \leq \frac{1}{\widehat{\mathrm{snr}}_i(d)}$. Hence, we can write an upper bound on $\epsilon$ as:

$$\begin{aligned}
\epsilon &\leq \frac{A+F}{\widehat{\mathrm{snr}}_i(d)} + \frac{D\sqrt{\log(k/\delta)}}{\widehat{\mathrm{snr}}_i(d)} + E\sqrt{\log(k/\delta)} \\
&\leq \frac{(A+F)(\sqrt{n^{(d)}} + \sqrt{T_0})}{\sigma_{r_d}(\mathbf{Z}_i(d))} + \frac{(A+F)\sqrt{\log(\log(n^{(d)})/\delta)}}{\sigma_{r_d}(\mathbf{Z}_i(d))} \\
&\quad + \frac{D\left(\sqrt{n} + \sqrt{d} + \sqrt{\log(\log(n^{(d)})/\delta)}\right)\sqrt{\log(k/\delta)}}{\sigma_r(\mathbf{Z}_i(d))} + E\sqrt{\log(k/\delta)}
\end{aligned}$$

Since $\log(x)$ is a strictly increasing function for $x > 0$, we can simplify the above expression as:

$$\begin{aligned}
\epsilon &\leq \frac{(A+F)(\sqrt{n^{(d)}} + \sqrt{T_0})}{\sigma_{r_d}(\mathbf{Z}_i(d))} + \frac{(A+F)\sqrt{\log\big(\log(n^{(d)}) \vee k/\delta\big)}}{\sigma_{r_d}(\mathbf{Z}_i(d))} \\
&\quad + \frac{D(\sqrt{n} + \sqrt{d})\sqrt{\log\big(\log(n^{(d)}) \vee k/\delta\big)}}{\sigma_{r_d}(\mathbf{Z}_i(d))} + \frac{D\log\big(\log(n^{(d)}) \vee k/\delta\big)}{\sigma_{r_d}(\mathbf{Z}_i(d))} + E\sqrt{\log\big(\log(n^{(d)}) \vee k/\delta\big)}
\end{aligned}$$

Subtracting the first term from both sides and multiplying by $\sigma_r(\mathbf{Z}_i(d))$, we have:

$$\begin{aligned}
&\sigma_{r_d}(\mathbf{Z}_i(d))\epsilon - (A+F)(\sqrt{n^{(d)}} + \sqrt{T_0}) \\
&\leq (A+F)\sqrt{\log\big(\log(n^{(d)}) \vee k/\delta\big)} + D(\sqrt{n} + \sqrt{d})\sqrt{\log\big(\log(n^{(d)}) \vee k/\delta\big)} + D\log\big(\log(n^{(d)}) \vee k/\delta\big) \\
&\quad + E\sigma_{r_d}(\mathbf{Z}_i(d))\sqrt{\log\big(\log(n^{(d)}) \vee k/\delta\big)} \\
&= (A + F + D(\sqrt{n} + \sqrt{d}) + E\sigma_{r_d}(\mathbf{Z}_i(d)))\sqrt{\log\big(\log(n^{(d)}) \vee k/\delta\big)} + D\log\big(\log(n^{(d)}) \vee k/\delta\big)
\end{aligned}$$

Let $\alpha = A + F + D(\sqrt{n} + \sqrt{d}) + E\sigma_{r_d}(\mathbf{Z}_i(d))$, we can rewrite the inequality above as:

$$\sigma_r(\mathbf{Z}_i(d))\epsilon - (A+F)(\sqrt{n^{(d)}} + \sqrt{T_0}) \leq \alpha\sqrt{\log\big(\log(n^{(d)}) \vee k/\delta\big)} + D\log\big(\log(n^{(d)}) \vee k/\delta\big)$$

Then, we can complete the square and obtain:

$$\log\big(\log(n^{(d)})\vee k/\delta\big) + \frac{\alpha}{D}\sqrt{\log\big(\log(n^{(d)})\vee k/\delta\big)} + \frac{\alpha^2}{4D^2} \geq \frac{\sigma_{r_d}(\mathbf{Z}_i(d))\epsilon - (A+F)(\sqrt{n^{(d)}} + \sqrt{T_0})}{D} + \frac{\alpha^2}{4D^2}$$

$$\iff \left(\sqrt{\log\big(\log(n^{(d)})\vee k/\delta\big)} + \frac{\alpha}{2D}\right)^2 \geq \frac{\sigma_{r_d}(\mathbf{Z}_i(d))\epsilon - (A+F)(\sqrt{n^{(d)}} + \sqrt{T_0})}{D} + \frac{\alpha^2}{4D^2}$$

$$\iff \sqrt{\log\big(\log(n^{(d)})\vee k/\delta\big)} + \frac{\alpha}{2D} \geq \sqrt{\frac{\sigma_{r_d}(\mathbf{Z}_i(d))\epsilon - (A+F)(\sqrt{n^{(d)}} + \sqrt{T_0})}{D} + \frac{\alpha^2}{4D^2}}$$

$$\iff \sqrt{\log\big(\log(n^{(d)})\vee k/\delta\big)} \geq \sqrt{\frac{\sigma_{r_d}(\mathbf{Z}_i(d))\epsilon - (A+F)(\sqrt{n^{(d)}} + \sqrt{T_0})}{D} + \frac{\alpha^2}{4C^2}} - \frac{\alpha}{2D}$$

$$\iff \log\big(\log(n^{(d)})\vee k/\delta\big) \geq \left(\sqrt{\frac{\sigma_{r_d}(\mathbf{Z}_i(d))\epsilon - (A+F)(\sqrt{n^{(d)}} + \sqrt{T_0})}{D} + \frac{\alpha^2}{4D^2}} - \frac{\alpha}{2D}\right)^2$$

$$\iff \frac{\log(n^{(d)})\vee k}{\delta} \geq \exp\left(\left(\sqrt{\frac{\sigma_{r_d}(\mathbf{Z}_i(d))\epsilon - (A+F)(\sqrt{n^{(d)}} + \sqrt{T_0})}{D} + \frac{\alpha^2}{4D^2}} - \frac{\alpha}{2D}\right)^2\right)$$

$$\iff \delta \leq \frac{\log(n^{(d)})\vee k}{\exp\left(\left(\sqrt{\frac{\sigma_{r_d}(\mathbf{Z}_i(d))\epsilon - (A+F)(\sqrt{n^{(d)}} + \sqrt{T_0})}{D} + \frac{\alpha^2}{4D^2}} - \frac{\alpha}{2D}\right)^2\right)}$$

$\square$

## C.2  Proof of Theorem 4.1

**Assumption C.3.** *[Unit Knowledge for Synthetic Control] We assume that the following are common knowledge among all units and the principal:*

1. *Valid upper- and lower-bounds on the fraction of type 1 units in the population, i.e. if the true proportion of type 1 units is $p_1 \in (0,1)$, the principal and units know $p_L, p_H \in (0,1)$ such that $p_L \leq p_1 \leq p_H$.*

2. *Valid upper- and lower-bounds on the prior mean for each receiver type and intervention, i.e. values $\mu^{(d,\tau,h)}, \mu^{(d,\tau,l)} \in \mathbb{R}$ such that $\mu^{(d,\tau,l)} \leq \mu_{v_i}^{(d)} \leq \mu^{(d,\tau,h)}$ for all units $i \in [\![N]\!]$, receiver types $\tau \in \{0,1\}$, and interventions $d \in \{0,1\}$.*

3. *For some $C \in (0,1)$, a lower bound on the smallest probability of the event $\xi_{C,i}$ over the priors and latent factors of type 1 units, denoted by $\min_{i \in \mathcal{T}^{(1)}} \Pr_{\mathcal{P}_i}[\xi_{C,i}] \geq \zeta_C > 0$, where $\xi_{C,i} := \left\{\mu_i^{(0)} \geq \bar{y}_{i,post}^{(1)} + C\right\}$ and $\bar{y}_{i,post}^{(1)}$ is the average post-intervention outcome for unit $i$ under intervention 1.*

4. *A sufficient number of observations $N_0$ needed such that the Unit Overlap Assumption is satisfied for type 1 units under the treatment with probability at least $1 - \delta$, for some $\delta \in (0,1)$.*

At a high level, part (3) requires that (i) there is a non-zero probability under each unit $i$'s prior that $\bar{y}_{i,post}^{(0)} \geq \bar{y}_{i,post}^{(1)}$ and (ii) the principal will be able to infer a lower bound on this probability given the set of observed outcomes for units in the first phase.[8]

---

[8]Such "fighting chance" assumptions are very common (and are necessary) in the literature on incentivized exploration in bandits.

**Validity of Assumption C.3.3** First, we note that in the first stage of Algorithm 1, the principal does not provide any recommendation to the units and instead lets them pick their preferred intervention. The goal of this first stage is to ensure the linear span inclusion assumption (Unit Overlap Assumption) is satisfied for type 1 units and intervention 1. This condition is equivalent to having enough samples of type 1 units such that the set of latent vectors $\{v_i\}_{i\in\mathcal{I}^{(1)}}$ spans the latent vector space $S_1$. We invoke the following theorem from Vershynin (2018) that shows $\mathrm{span}(\{v_i\}_{i\in\mathcal{I}^{(1)}}) = S_1$ with high probability:

**Theorem C.4** (Theorem 4.6.1 of Vershynin (2018)). *Let $A$ be an $m \times n$ matrix whose rows $A_i$ are independent, mean zero, sub-gaussian isotropic random vectors in $\mathbb{R}^n$. Then for any $t \geq 0$ we have with probability at least $1 - 2\exp(-t^2)$:*

$$\sqrt{m} - c_{Ver}K^2(\sqrt{n} + t) \leq s_n(A) \leq s_1(A) \leq \sqrt{m} + c_{Ver}K^2(\sqrt{n} + t) \tag{1}$$

*where $K = \max_i \|A_i\|_{\psi_2}$ and $c_{Ver}$ is an absolute constant.*

Hence, after observing $N_0^{(1)}$ samples of type 1 units taking intervention 1, the linear span inclusion assumption is satisfied with probability at least $1 - 2\exp\left(-\left(\frac{\sqrt{N_0^{(1)}}}{c_{Ver}K^2} - \sqrt{r_1}\right)^2\right)$.

**Theorem C.5.** *Suppose the there are two interventions, and assume that Assumption C.3 holds for some constant gap $C \in (0,1)$. If $N_0$ is chosen to be large enough such that Unit Overlap Assumption is satisfied for all units of type 1 under treatment with probability at least $1 - \delta_0$, then Algorithm 1 with parameters $\delta, B, C$ is BIC for all units of type 1 if*

$$L \geq 1 + \max_{i \in \mathcal{T}^{(1)}} \left\{ \frac{\mu_{v_i}^{(1)} - \mu_{v_i}^{(0)}}{\left(C - \alpha(\delta_{PCR}) - \sigma\sqrt{\frac{2\log(1/\delta_{\epsilon_i})}{T_1}}\right)\Pr\left[\mu_{v_i}^{(0)} - \bar{y}_{i,post}^{(1)} \geq C - \alpha(\delta_{PCR}) - \sigma\sqrt{\frac{2\log(1/\delta_{\epsilon_i})}{T_1}}\right] - 2\delta} \right\}$$

*where $\alpha(\delta_{PCR})$ is the high-probability confidence bound defined in Theorem C.1, $\delta = \delta_{\epsilon_i} + \delta_0 + \delta_{PCR}$ and $\delta_{\epsilon_i} \in (0,1)$ is the failure probability of the Chernoff-Hoeffding bound on the average of sub-Gaussian random noise $\{\epsilon_{i,t}^{(1)}\}_{t=T_0+1}^T$, and*

$$\delta_{PCR} \leq \frac{\log(N_0^{(1)}) \vee k}{\exp\left(\left(\sqrt{\frac{\sigma_r(Y_{pre,N_0}^{(1)})(C/2)-(A+F)(\sqrt{N_0^{(1)}}+\sqrt{T_0})}{D} + \frac{\alpha^2}{4D^2}} - \frac{\alpha}{2D}\right)^2\right)}$$

*and $\kappa\left(Y_{pre,N_0}^{(1)}\right) = \sigma_1\left(Y_{pre,N_0}^{(1)}\right)/\sigma_{r_1}\left(Y_{pre,N_0}^{(1)}\right)$ is the condition number of the matrix of observed pre-intervention outcomes for the subset of the first $N_0$ units who have taken the treatment. The remaining variables in $\delta_{PCR}$ are defined as $A = 3\sqrt{T_0}\left(\frac{\Gamma(\sqrt{74}+12\sqrt{6}\kappa(Y_{pre,N_0}^{(1)}))}{T-T_0} + \frac{\sqrt{2}}{\sqrt{T-T_0}}\right)$, $F = \frac{2\Gamma\sqrt{24T_0}}{T-T_0} + \frac{12\Gamma\kappa(Y_{pre,N_0}^{(1)})\sqrt{3T_0}}{T-T_0} + \frac{2}{\sqrt{T-T_0}}$, $D = \frac{\Gamma\sigma\sqrt{148}}{T-T_0} + \frac{24\sigma\kappa(Y_{pre,N_0}^{(1)})\sqrt{6}}{T-T_0} + \frac{2\sigma}{\sqrt{T-T_0}}$, $E = \frac{\sqrt{2}\Gamma\sigma}{T-T_0}$, and $\alpha = A + F + D(\sqrt{N_0^{(1)}} + \sqrt{T_0}) + \sigma_{r_1}(Y_{pre,N_0}^{(1)})E$. Moreover if the number of batches $B$ is chosen to be large enough such that with probability at least $1 - \delta$, $\mathrm{rank}([\mathbb{E}[\mathbf{y}_{i,pre}]^\top : \hat{d}_i = 0, i \in \mathcal{T}^{(1)}]_{i=1}^{N_0+B\cdot L}) = \mathrm{rank}([\mathbb{E}[\mathbf{y}_{j,pre}]^\top]_{j\in\mathcal{T}^{(1)}})$, then the Unit Overlap Assumption will be satisfied for all type 1 units under control with probability at least $1 - 2\delta$.*

*Proof.* At a particular time step $t$, unit $i$ of type 1 can be convinced to pick control if $\mathbb{E}_{v_i}[\bar{y}_{i,post}^{(0)} - \bar{y}_{i,post}^{(1)}|\hat{d} = 0]\Pr[\hat{d} = 0] \geq 0$. There are two possible disjoint events under which unit $i$ is recommended intervention 0: either intervention 0 is better empirically, i.e. $\mu_{v_i}^{(0)} \geq \widehat{\mathbb{E}}[\bar{y}_{i,post}^{(1)}] + C$, or intervention 1 is better and unit $i$ is

chosen for exploration. Hence, we have

$$\mathbb{E}_{v_i}[\bar{y}^{(0)}_{i,post} - \bar{y}^{(1)}_{i,post}|\widehat{d} = 0]\Pr[\widehat{d} = 0]$$

$$= \mathbb{E}[\bar{y}^{(0)}_{i,post} - \bar{y}^{(1)}_{i,post}|\mu^{(0)}_{v_i} \geq \widehat{\mathbb{E}}[\bar{y}^{(1)}_{i,post}] + C]\Pr[\mu^{(0)}_{v_i} \geq \widehat{\mathbb{E}}[\bar{y}^{(1)}_{i,post}] + C]\left(1 - \frac{1}{L}\right) + \frac{1}{L}\mathbb{E}_{v_i}[\bar{y}^{(0)}_{i,post} - \bar{y}^{(1)}_{i,post}]$$

$$= \left(\mu^{0}_{v_i} - \mathbb{E}_{v_i}[\bar{y}^{(1)}_{i,post}|\mu^{(0)}_{v_i} \geq \widehat{\mathbb{E}}[\bar{y}^{(1)}_{i,post}] + C]\right)\Pr[\mu^{(0)}_{v_i} \geq \widehat{\mathbb{E}}[\bar{y}^{(1)}_{i,post}] + C]\left(1 - \frac{1}{L}\right) + \frac{1}{L}(\mu^{(0)}_{v_i} - \mu^{(1)}_{v_i})$$

Rearranging the terms and taking the maximum over all units of type 1 gives the lower bound on phase length $L$:

$$L \geq 1 + \frac{\mu^{(1)}_{v_i} - \mu^{(0)}_{v_i}}{(\mu^{(0)}_{v_i} - \mathbb{E}_{v_i}[\widehat{\mathbb{E}}[\bar{y}^{(1)}_{i,post}]|\widehat{\xi}_{C,i}])\Pr_{v_i}[\widehat{\xi}_{C,i}]}$$

To complete the analysis, we want to find a lower bound for the terms in the denominator. That is, we want to lower bound

$$\left(\mu^{(0)}_{v_i} - \mathbb{E}_{v_i}[\bar{y}^{(1)}_{i,post}|\mu^{(0)}_{v_i} \geq \widehat{\mathbb{E}}[\bar{y}^{(1)}_{i,post}] + C]\right)\Pr_{v_i}[\mu^{(0)}_{v_i} \geq \widehat{\mathbb{E}}[\bar{y}^{(1)}_{i,post}] + C]$$

Let $\xi_0$ denote the event that Unit Overlap Assumption is satisfied for type 1 units under treatment. From Assumption C.3, we know that event $\xi_0$ occurs with probability at least $1 - \delta_0$ for some $\delta_0 \in (0, 1)$. Let $\xi_{PCR}$ denote the event that the high-probability concentration bound holds for units of type 1 and intervention 1. Finally, let $\xi_{\epsilon_i}$ denote the event where the Chernoff-Hoeffding bound holds on the average of the sub-Gaussian noise $\epsilon^{(1)}_{i,t}$, that is with probability at least $1 - \delta_{\epsilon_i}$, we have:

$$\left|\frac{1}{T_1}\sum_{t=T_0+1}^{T}\epsilon^{(1)}_{i,t}\right| \leq \sigma\sqrt{\frac{2\log(1/\delta_{\epsilon_i})}{T_1}}$$

Define another clean event $\xi$ where the events $\xi_0$, $\xi_{\epsilon_i}$ and $\xi_{PCR}$ happen simultaneously with probability at least $1 - \delta$, where $\delta = \delta_0 + \delta_{PCR} + \delta_{\epsilon_i}$. Then, we have:

$$(\mu^{(0)}_{v_i} - \mathbb{E}_{v_i}[\bar{y}^{(1)}_{i,post}|\widehat{\xi}_{C,i}])\Pr[\widehat{\xi}_{C,i}]$$

$$= (\mu^{(0)}_{v_i} - \mathbb{E}_{v_i}[\bar{y}^{(1)}_{i,post}|\widehat{\xi}_{C,i}, \xi])\Pr[\widehat{\xi}_{C,i}, \xi] + (\mu^{(0)}_{v_i} - \mathbb{E}_{v_i}[\bar{y}^{(1)}_{i,post}|\widehat{\xi}_{C,i}, \neg\xi])\Pr[\widehat{\xi}_{C,i}, \neg\xi]$$

$$\geq (\mu^{(0)}_{v_i} - \mathbb{E}_{v_i}[\bar{y}^{(1)}_{i,post}|\widehat{\xi}_{C,i}, \xi])\Pr[\widehat{\xi}_{C,i}, \xi] - 2\delta \qquad \text{(since } \Pr[\neg\xi] < \delta \text{ and } \mu^{(0)}_{v_i} - \mathbb{E}_{v_i}[\bar{y}^{(1)}_{i,post}] \geq -2)$$

We proceed to lower bound the expression above as follows:

$$\mu^{(0)}_{v_i} - \mathbb{E}_{v_i}[\bar{y}^{(1)}_{i,post}|\widehat{\xi}_{C,i}, \xi]$$

$$= \mu^{(0)}_{v_i} - \mathbb{E}_{v_i}\left[\bar{y}^{(1)}_{i,post}|\mu^{(0)}_{v_i} \geq \widehat{\mathbb{E}}[\bar{y}^{(1)}_{i,post} + C], \left|\widehat{\mathbb{E}}[\bar{y}^{(1)}_{i,post}] - \mathbb{E}[\bar{y}^{(1)}_{i,post}]\right| \leq \alpha(\delta_{PCR}), \left|\frac{1}{T_1}\sum_{t=T_0+1}^{T}\epsilon^{(1)}_{i,t}\right| \leq \sigma\sqrt{\frac{2\log(1/\delta_{\epsilon_i})}{T_1}}\right]$$

$$= \mu^{(0)}_{v_i} - \mathbb{E}_{v_i}\left[\bar{y}^{(1)}_{i,post}|\mu^{(0)}_{v_i} - \mathbb{E}[\bar{y}^{(1)}_{i,post}] \geq C - \alpha(\delta_{PCR}), \left|\frac{1}{T_1}\sum_{t=T_0+1}^{T}\epsilon^{(1)}_{i,t}\right| \leq \sigma\sqrt{\frac{2\log(1/\delta_{\epsilon_i})}{T_1}}\right]$$

$$= \mu^{(0)}_{v_i} - \mathbb{E}[v_i]\left[\bar{y}^{(1)}_{i,post}|\mu^{(0)}_{v_i} - \bar{y}^{(1)}_{i,post} \geq C - \alpha(\delta_{PCR}) - \sigma\sqrt{\frac{2\log(1/\delta_{\epsilon_i})}{T_1}}\right]$$

$$\geq C - \alpha(\delta_{PCR}) - \sigma\sqrt{\frac{2\log(1/\delta_{\epsilon_i})}{T_1}}$$

Furthermore, we can write the probability of the joint event $\widehat{\xi}_{C,i}, \xi$ as:

$$
\begin{aligned}
&\Pr[\widehat{\xi}_{C,i}, \xi] \\
&= \Pr\left[\mu_{v_i}^{(0)} \geq \widehat{\mathbb{E}}[\bar{y}_{i,post}^{(1)} + C], \left|\widehat{\mathbb{E}}[\bar{y}_{i,post}^{(1)}] - \mathbb{E}[\bar{y}_{i,post}^{(1)}]\right| \leq \alpha(\delta_{PCR}), \left|\frac{1}{T_1} \sum_{t=T_0+1}^{T} \epsilon_{i,t}^{(1)}\right| \leq \sigma\sqrt{\frac{2\log(1/\delta_{\epsilon_i})}{T_1}}\right] \\
&= \Pr\left[\mu_{v_i}^{(0)} - \bar{y}_{i,post}^{(1)} \geq C - \alpha(\delta_{PCR}) - \sigma\sqrt{\frac{2\log(1/\delta_{\epsilon_i})}{T_1}}\right]
\end{aligned}
$$

Hence, we can derive the following lower bound on the denominator of $L$:

$$
\begin{aligned}
&\mu_{v_i}^{(0)} - \mathbb{E}_{v_i}[\bar{y}_{i,post}^{(1)} | \widehat{\xi}_{C,i}] \\
&\geq \left(C - \alpha(\delta_{PCR}) - \sigma\sqrt{\frac{2\log(1/\delta_{\epsilon_i})}{T_1}}\right) \Pr\left[\mu_{v_i}^{(0)} - \bar{y}_{i,post}^{(1)} \geq C - \alpha(\delta_{PCR}) - \sigma\sqrt{\frac{2\log(1/\delta_{\epsilon_i})}{T_1}}\right] - 2\delta
\end{aligned}
$$

Applying this lower bound to the expression of $L$ and taking the maximum over type 1 units, we have:

$$
L \geq 1 + \max_{i \in \mathcal{I}^{(1)}} \left\{\frac{\mu_{v_i}^{(1)} - \mu_{v_i}^{(0)}}{\left(C - \alpha(\delta_{PCR}) - \sigma\sqrt{\frac{2\log(1/\delta_{\epsilon_i})}{T_1}}\right) \Pr\left[\mu_{v_i}^{(0)} - \bar{y}_{i,post}^{(1)} \geq C - \alpha(\delta_{PCR}) - \sigma\sqrt{\frac{2\log(1/\delta_{\epsilon_i})}{T_1}}\right] - 2\delta}\right\}
$$

$\square$

**Example C.6.** *Consider the following data-generating process: Suppose there are two receiver types, each with equal probability; type 1 units prefer the treatment and type 0 units prefer control. Let $v_i \sim \text{Unif}[0.25, 0.75]$. If unit $i$ is a type 1 unit, they have latent factor $\mathbf{v}_i = [v_i\ 0]$. Otherwise they have latent factor $\mathbf{v}_i = [0\ v_i]$. Suppose that the pre-intervention period is of length $T_0 = 2$ and the post-intervention period is of length $T_1 = 1$. We leave $\mathbf{u}_1^{(0)}, \mathbf{u}_2^{(0)}, \mathbf{u}_3^{(0)}, \mathbf{u}_3^{(1)}$ unspecified.*

*Since $r_1 = 1$, we only need to incentivize a single type 1 unit to take the control in order for the Unit Overlap Assumption to be satisfied for type 1 units under control. Therefore, for a given confidence level $\delta$, it suffices to set $N_0 = B = \frac{\log(1/\delta)}{\log(2)}$. After the first $N_0$ time-steps, the principal will know all of the parameters necessary to compute the batch size $L$.*

*Since $T_1 = 1$, unit $i$'s prior is over $\{\mathbb{E}[y_{i,3}^{(0)}], \mathbb{E}[y_{i,3}^{(1)}]\}$. For simplicity, suppose that (1) all type 1 units believe that $\mathbb{E}[y_{i,3}^{(0)}] \sim \text{Unif}[0, 0.5]$ and $\mathbb{E}[y_{i,3}^{(1)}] \sim \text{Unif}[0, 1]$ and (2) there is no noise in the post-intervention outcome. Under this setting, the event $\xi_{C,i}$ simplifies to $\xi_{C,i} = \left\{0.25 \leq \mathbb{E}[y_{i,3}^{(1)}] + C\right\}$ for any unit $i$ and constant $C \in (0, 1)$. Therefore*

$$
\zeta_C = \Pr_{\mathcal{P}_i}[\mathbb{E}[y_{i,3}^{(1)}] \leq 0.25 - C] = \max\{0, 0.25 - C\}.
$$

# D Extension to synthetic interventions

Our focus so far has been to incentivize units who *a priori* prefer the (single) treatment to take the control in order to obtain accurate counterfactual estimates for all units under control. In this section, we extend Algorithm 1 to the setting where there are *multiple* treatments. Now our goal is to incentivize enough exploration across all interventions such that the Unit Overlap Assumption is satisfied *for every intervention* and thus our counterfactual estimates are valid simultaneously for all units in the population under every intervention. In order to do so, we use tools from the literature on *synthetic interventions*, which is a generalization of the synthetic control framework to allows for counterfactual estimation under different treatments, in addition to control Agarwal et al. (2020b; 2023).

---

**ALGORITHM 2:** Incentivizing Exploration for Synthetic Interventions: sub-type $\boldsymbol{\tau}$ units

---

**Input:** Group size $L$, number of batches $B$, failure probability $\delta$, fixed gap $C \in (0, 1)$

Provide no recommendation to first $N_0$ units.

`// Loop through explore interventions`

**for** $\ell = \boldsymbol{\tau}[k], \boldsymbol{\tau}[k-1], \ldots, \boldsymbol{\tau}[2]$ **do**

    **for** *batch* $b = 1, 2, \cdots, B$ **do**

        Select an explore index $i_b \in [L]$ uniformly at random.

        **for** $j = 1, 2, \cdots, L$ **do**

            **if** $j = i_b$ **then**

                Recommend intervention $\widehat{d}_{N_0 + (b-1) \cdot L + j} = \ell$

            **else**

                **if** $\widehat{\mathbb{E}}[\bar{y}_{i,post}^{(\boldsymbol{\tau}[1])}] + C < \widehat{\mathbb{E}}[\bar{y}_{i,post}^{(d)}] < \mu^{(\ell, \boldsymbol{\tau}, l)} - C$ *for every intervention* $d \in \{\ell - 1, \ldots, \boldsymbol{\tau}[k]\}$ **then**

                    Recommend intervention $\widehat{d}_{N_0 + (b-1) \cdot L + j} = \ell$

                **else**

                    Recommend intervention $\widehat{d}_{N_0 + (b-1) \cdot L + j} = \boldsymbol{\tau}[1]$

---

Our setup is the same as in Section 4, with the only difference being that each unit may choose one of $k \geq 2$ interventions after the pre-treatment period. We assume that Assumption 2.1, Definition 2.4, Definition 2.5, and Assumption 3.2 are all extended to hold under $k$ interventions. Under this setting, a unit's beliefs induce a preference ordering over different interventions. We capture this through the notion of a unit *sub-type*, which is a generalization of our definition of unit type to the setting with $k$ interventions.

**Definition D.1** (Unit Sub-type). *A unit sub-type $\boldsymbol{\tau} \in [k]_0^k$ is a preference ordering over interventions. Unit $i$ is of sub-type $\boldsymbol{\tau}$ if for every $\kappa \leq k$,*

$$\boldsymbol{\tau}[\kappa] = \arg \max_{d \in [\![k]\!]_0 \setminus \{\boldsymbol{\tau}[\kappa']\}_{\kappa' < \kappa}} \mu_{v_i}^{(d)}.$$

*Unit $i$'s type is $\boldsymbol{\tau}[1]$. We denote the set of all units of sub-type $\boldsymbol{\tau}$ as $\mathcal{T}^{(\boldsymbol{\tau})}$ and the dimension of the latent subspace spanned by sub-type $\boldsymbol{\tau}$ units as $r_{\boldsymbol{\tau}} := \mathrm{span}(\mathbf{v}_j : j \in \mathcal{T}^{(\boldsymbol{\tau})})$.*

Intuitively, a unit sub-type $\boldsymbol{\tau}$ is just the preference ordering over interventions such that intervention $\boldsymbol{\tau}[a]$ is preferred to intervention $\boldsymbol{\tau}[b]$ for $a < b$.[9] Since different unit sub-types have different preference orderings over interventions, we design our algorithm (Algorithm 2) to be BIC for all units of a given sub-type (in contrast to Algorithm 1, which is BIC for all units of a given type).

While there can be up to $k!$ possible sub-types, recall that the unit latent space has dimension $r$. Therefore, there can be at most $r$ linearly independent subspaces that matter for overlap. As a result, if we want to satisfy overlap for all $k!$ subtypes, we only need to run Algorithm 2 for enough unit subtypes to span this $r$ dimensional space. [10]

For a given sub-type $\boldsymbol{\tau}$, the goal of Algorithm 2 is to incentivize sufficient exploration for the Unit Overlap Assumption to be satisfied for all interventions with high probability. As was the case in Section 4, the algorithm will still be split into two phases, and units will not be given any recommendation in the first phase. The main difference compared to Algorithm 1 is how the principal chooses the exploit intervention in the second phase.

When there are only two interventions, the principal can compare their estimate of the average counterfactual outcome under treatment to the lower bound on the prior-mean average counterfactual outcome for control for type 1 units (and vice versa for type 0 units). However, a more complicated procedure is required to maintain Bayesian incentive-compatibility when there are more than two interventions.

---

[9]Note that it is without loss of generality to assume that units have a preference ordering over the interventions, as long as any tie-breaking is done in a known and deterministic way.

[10]This is because if the Unit Overlap Assumption is satisfied for a specific unit sub-type whose latent factors lie within a particular subspace, it is also satisfied for all other units whose latent factors lie within the same subspace, and there are of course at most $r$ different subspaces by Assumption 2.1.

Instead, in Algorithm 2, the principal will set the exploit intervention as follows: For receiver sub-type $\boldsymbol{\tau}$, Algorithm 2 sets intervention $\ell$ to be the exploit intervention for unit $i$ only if the sample average of *all* interventions $d \in \{\ell-1, \ldots, \boldsymbol{\tau}[k]\}$ are both (1) larger than the sample average of intervention $\boldsymbol{\tau}[1]$ by some constant gap $C$ and (2) less than the lower bound on the prior-mean average counterfactual outcome $\mu_i^{(\ell)}$ by $C$. If no such intervention satisfies both conditions, intervention $\boldsymbol{\tau}[1]$ is chosen to be the exploit intervention.

We require the following "common knowledge" assumption for Algorithm 2, which is analogous to Assumption C.3:

**Assumption D.2.** *We assume that the following are common knowledge among all units and the principal:*

1. *Valid upper- and lower-bounds on the fraction of sub-type $\boldsymbol{\tau}$ units in the population, i.e. if the true proportion of sub-type $\boldsymbol{\tau}$ units is $p_{\boldsymbol{\tau}} \in (0,1)$, the principal knows $p_{\boldsymbol{\tau},L}, p_{\boldsymbol{\tau},H} \in (0,1)$ such that*

$$p_{\boldsymbol{\tau},L} \leq p_{\boldsymbol{\tau}} \leq p_{\boldsymbol{\tau},H}.$$

2. *Valid upper- and lower-bounds on the prior mean for each receiver sub-type and intervention, i.e. values $\mu^{(d,\boldsymbol{\tau},h)}, \mu^{(d,\boldsymbol{\tau},l)} \in \mathbb{R}$ such that*

$$\mu^{(d,\boldsymbol{\tau},l)} \leq \mu_i^{(d)} \leq \mu^{(d,\boldsymbol{\tau},h)}$$

   *for all receiver sub-types $\boldsymbol{\tau}$, interventions $d \in [\![k]\!]_0$ and all units $i \in \mathcal{T}^{(\boldsymbol{\tau})}$.*

3. *For some $C \in (0,1)$, a lower bound on the smallest probability of the event $\mathcal{E}_{C,i}$ over the prior and latent factors of sub-type $\boldsymbol{\tau}$ units, denoted by $\min_{i \in \mathcal{T}^{(\tau)}} \Pr_{\mathcal{P}_i}[\mathcal{E}_{C,i}] \geq \zeta_C > 0$, where*

$$\mathcal{E}_{C,i} := \{\forall j \in [\![k]\!]_0 \backslash \{\boldsymbol{\tau}[1], \boldsymbol{\tau}[k]\} : \bar{y}_{i,post}^{(\boldsymbol{\tau}[1])} + C \leq \bar{y}_{i,post}^{(j)} \leq \mu_i^{(\boldsymbol{\tau}[k])} - C\}$$

   *and $\bar{y}_{i,post}^{(\ell)}$ is the average post-intervention outcome for unit $i$ under intervention $\ell \in [\![k]\!]_0$.*

4. *A sufficient number of observations $N_0$ needed such that the Unit Overlap Assumption is satisfied for sub-type $\boldsymbol{\tau}$ units under intervention $\boldsymbol{\tau}[1]$ with probability at least $1 - \delta$, for some $\delta \in (0,1)$.*

**Theorem D.3** (Informal; detailed version in Theorem E.1)**.** *Under Assumption D.2, if the number of initial units $N_0$ and batch size $L$ are chosen to be sufficiently large, then Algorithm 2 satisfies the BIC property (Definition 2.7).*

*Moreover, if the number of batches $B$ is chosen to be sufficiently large, then the Unit Overlap Assumption will be satisfied for all units of sub-type $\boldsymbol{\tau}$ under all interventions with high probability.*

See Appendix E for the proof. The main difference compared to the analysis of Algorithm 1 is in the definition of the "exploit" intervention: First, when there are $k$ interventions, an intervention $\ell$ is chosen as the exploit intervention only when the sample average of each intervention $d \in \{\ell-1, \ldots, \boldsymbol{\tau}[k]\}$ is (i) larger than the sample average of intervention $\boldsymbol{\tau}[1]$, and (ii) smaller than the prior-mean average outcome of intervention $\ell$ (with some margin $C$). Second, instead of choosing between a "previously best" intervention (i.e., an intervention $\boldsymbol{\tau}[j] < \ell$ with the highest sample average) and intervention $\ell$ to be the 'exploit' intervention, Algorithm 2 always choose between $\boldsymbol{\tau}[1]$ and $\ell$. This is because in the former case, conditional on any given intervention being chosen as the exploit intervention, it is an open technical challenge to show that the exploit intervention will have a higher average expected outcome compared to all other interventions.[11]

# E   Appendix for Section D: Extension to synthetic interventions

**Theorem E.1.** *Suppose that Assumption D.2 holds for some constant gap $C \in (0,1)$. If the number of initial units $N_0$ is chosen to be large enough such that Unit Overlap Assumption is satisfied for all units of*

---

[11]This observation is in line with work on frequentist-based BIC algorithms for incentivizing exploration in bandits. See Section 6.2 of Mansour et al. (2020) for more details.

*subtype $\boldsymbol{\tau}$ under intervention $\boldsymbol{\tau}[1]$ with probability $1 - \delta$, then Algorithm 2 with parameters $\delta, L, B, C$ is BIC for all units of subtype $\boldsymbol{\tau}$ if:*

$$L \geq 1 + \max_{i \in \mathcal{T}^{(\tau)}} \left\{ \frac{\mu_i^{(\tau[1])} - \mu_i^{(\tau[k])}}{\left( C - 2\alpha(\delta_{PCR}) - 2\sigma\sqrt{\frac{2\log(1/\delta_{\epsilon_i})}{T_1}} \right) (1 - \delta) \Pr\left[ \mathcal{E}_{C,i} \right] - 2\delta} \right\}$$

*Moreover, if the number of batches $B$ is chosen to be large enough such that with probability at least $1 - \delta$, we have* $\mathrm{rank}([\mathbb{E}[\mathbf{y}_{i,pre}]^\top : \widehat{d}_i = \ell \neq \boldsymbol{\tau}[1], i \in \mathcal{T}^{(\tau)}]_{i=1}^{N_0 + B \cdot L}) = \mathrm{rank}([\mathbb{E}[\mathbf{y}_{j,pre}]^\top]_{j \in \mathcal{T}^{(\tau)}})$, *then the Unit Overlap Assumption will be satisfied for all units of subtype $\boldsymbol{\tau}$ under all interventions with probability at least $1 - \mathcal{O}(k\delta)$.*

*Proof.* According to our recommendation policy, unit $i$ is either recommended intervention $\boldsymbol{\tau}[1]$ or intervention $\ell$. We will prove that in either case, unit $i$ will comply with the principal's recommendation.

Let $\widehat{\mathcal{E}}_{C,i}^{(\ell)}$ denote the event that the exploit intervention for unit $i$ is intervention $\ell$. Formally, we have

$$\widehat{\mathcal{E}}_{C,i}^{(\ell)} = \left\{ \widehat{\mathbb{E}}[\bar{y}_{i,post}^{(\tau[1])}] \leq \min_{1 < j < \ell} \widehat{\mathbb{E}}[\bar{y}_{i,post}^{(\tau[j])}] - C \quad \text{and} \quad \max_{1 \leq j < \ell} \widehat{\mathbb{E}}[\bar{y}_{i,post}^{(\tau[j])}] \leq \mu_{v_i}^{(\ell)} - C \right\}$$

**When unit $i$ is recommended intervention $\ell$:** When a unit $i \in \mathcal{T}^{(\tau)}$ is recommended intervention $\ell$, we argue that this unit will not switch to any other intervention $j \neq \ell$. Because of our ordering of the prior mean reward, any intervention $j > \ell$ has had no sample collected by a type $\boldsymbol{\tau}$ unit and $\mu_{v_i}^{(\tau[j])} \leq \mu_{v_i}^{(\tau[\ell])}$. Hence, we only need to focus on the cases where $j < \ell$. For the recommendation policy to be BIC, we need to show that

$$\mathbb{E}[\mu_{v_i}^{(\ell)} - \bar{y}_{i,post}^{(\tau[j])} | \widehat{d} = \ell] \Pr[\widehat{d} = \ell] \geq 0$$

There are two possible disjoint events under which unit $i$ is recommended intervention $\ell$: either $\ell$ is determined to be the 'exploit' intervention or unit $i$ is chosen as an 'explore' unit. Since being chosen as an explore unit does not imply any information about the rewards, we can derive that conditional on being in an 'explore' unit, the expected gain for unit $i$ to switch to intervention $j$ is simply $\mu_{v_i}^{(\ell)} - \mu_{v_i}^{(\tau[j])}$. On the other hand, if intervention $\ell$ is the 'exploit' intervention, then event $\widehat{\mathcal{E}}_{C,i}^{(\ell)}$ has happened. Hence, we can rewrite the left-hand side of the BIC condition above as:

$$\mathbb{E}[\mu_{v_i}^{(\ell)} - \bar{y}_{i,post}^{(\tau[j])} | \widehat{d} = \ell] \Pr[\widehat{d} = \ell] = \mathbb{E}[\mu_{v_i}^{(\ell)} - \bar{y}_{i,post}^{(\tau[j])} | \widehat{\mathcal{E}}_{C,i}^{(\ell)}] \Pr[\widehat{\mathcal{E}}_{C,i}^{(\ell)}] \left( 1 - \frac{1}{L} \right) + \frac{1}{L} (\mu_{v_i}^{(\ell)} - \mu_{v_i}^{(j)})$$

Rearranging the terms, we have the following lower bound on the phase length $L$ for the algorithm to be BIC for type 1 units:

$$L \geq 1 + \frac{\mu_{v_i}^{(\tau[j])} - \mu_{v_i}^{(\ell)}}{\mathbb{E}[\mu_{v_i}^{(\ell)} - \bar{y}_{i,post}^{(\tau[j])} | \widehat{\mathcal{E}}_{C,i}^{(\ell)}] \Pr[\widehat{\mathcal{E}}_{C,i}^{(\ell)}]}$$

To complete the analysis, we need to lower bound the denominator of the expression above. Consider the following event $\mathcal{C}$:

$$\mathcal{C} = \{ \forall j \in [k] : \left| \widehat{\mathbb{E}}[\bar{y}_i^{(\tau[j])}] - \mathbb{E}[\bar{y}_{i,post}^{(\tau[j])}] \right| \leq \alpha(\delta_{PCR}) \}$$

Let $\mathcal{E}_0$ denote the event that Unit Overlap Assumption is satisfied for units of type $\boldsymbol{\tau}$ under intervention $\boldsymbol{\tau}[1]$. From Assumption D.2, we know that event $\mathcal{E}_0$ occurs with probability at least $1 - \delta_0$ for some $\delta_0 \in (0, 1)$. Furthermore, let $\mathcal{E}_{\epsilon_i}$ denote the event where the Chernoff-Hoeffding bound on the noise sequences $\{\epsilon_{i,t}^{(\tau[j])}\}_{t=T_0+1}^{T}$ holds, that is with probability at least $1 - \delta_{\epsilon_i}$, we have:

$$\left| \frac{1}{T_1} \sum_{t=T_0+1}^{T} \epsilon_{i,t}^{(\tau[j])} \right| \leq \sigma \sqrt{\frac{2\log(1/\delta_{\epsilon_i})}{T_1}}$$

Then, we can define a clean event $\mathcal{E}$ where the events $\mathcal{C}$, $\mathcal{E}_0$ and $\mathcal{E}_{\epsilon_i}$ happens simultaneously with probability at least $1 - \delta$, where $\delta = \delta_\epsilon^{PCR} + \delta_{\epsilon_i} + \delta_0$.

Note that since $\left| \bar{y}_{i,post}^{(\tau[j])} \right| \le 1$, we can rewrite the denominator as:

$$\mathbb{E}[\mu_{v_i}^{(\ell)} - \bar{y}_{i,post}^{(\tau[j])} | \widehat{\mathcal{E}}_{C,i}] \Pr[\widehat{\mathcal{E}}_{C,i}^{(\ell)}]$$
$$= \mathbb{E}[\mu_{v_i}^{(\ell)} - \bar{y}_{i,post}^{(\tau[j])} | \widehat{\mathcal{E}}_{C,i}^{(\ell)}, \mathcal{E}] \Pr[\widehat{\mathcal{E}}_{C,i}^{(\ell)}, \mathcal{E}] + \mathbb{E}[\mu_{v_i}^{(\ell)} - \bar{y}_{i,post}^{(\tau[j])} | \widehat{\mathcal{E}}_{C,i}^{(\ell)}, \neg\mathcal{E}] \Pr[\widehat{\mathcal{E}}_{C,i}^{(\ell)}, \neg\mathcal{E}]$$
$$\ge \mathbb{E}[\mu_{v_i}^{(\ell)} - \bar{y}_{i,post}^{(\tau[j])} | \widehat{\mathcal{E}}_{C,i}^{(\ell)}, \mathcal{E}] \Pr[\widehat{\mathcal{E}}_{C,i}^{(\ell)}, \mathcal{E}] - 2\delta$$

Define an event $\mathcal{E}_i^{(\ell)}$ on the true average expected post-intervention outcome of each intervention as follows:

$$\mathcal{E}_i^{(\ell)} := \left\{ \bar{y}_{i,post}^{(\tau[1])} \le \min_{1 < j < \ell} \bar{y}_{i,post}^{(\tau[j])} - C - 2\alpha(\delta_{PCR}) - 2\sigma\sqrt{\frac{2\log(1/\delta_{\epsilon_i})}{T_1}} \right.$$

$$\left. \text{and} \quad \max_{1 \le j < \ell} \bar{y}_{i,post}^{(\tau[j])} \le \mu_{v_i}^{(\ell)} - C - 2\alpha(\delta_{PCR}) - 2\sigma\sqrt{\frac{2\log(1/\delta_{\epsilon_i})}{T_1}} \right\}$$

We can observe that under event $\mathcal{E}$, event $\widehat{\mathcal{E}}_{C,i}^{(\ell)}$ is implied by event $\mathcal{E}_i^{(\ell)}$. Hence, we have:

$$\Pr[\mathcal{E}, \mathcal{E}_i^{(\ell)}] \le \Pr[\mathcal{E}, \widehat{\mathcal{E}}_{C,i}^{(\ell)}]$$

We can rewrite the left-hand side as:

$$\Pr[\mathcal{C}, \mathcal{E}_i] = \Pr[\mathcal{C} | \mathcal{E}_i^{(\ell)}] \Pr[\mathcal{E}_i^{(\ell)}] \ge (1 - \delta) \Pr[\mathcal{E}_i^{(\ell)}]$$

Substituting these expressions using $\mathcal{E}_i^{(\ell)}$ for the ones using $\widehat{\mathcal{E}}_{C,i}$ in the denominator gives:

$$\mathbb{E}[\mu_{v_i}^{(\ell)} - \bar{y}_{i,post}^{(\tau[j])} | \widehat{\mathcal{E}}_{C,i}^{(\ell)}] \Pr[\widehat{\mathcal{E}}_{C,i}^{(\ell)}] \ge \left( C - 2\alpha(\delta_{PCR}) - 2\sigma\sqrt{\frac{2\log(1/\delta_{\epsilon_i})}{T_1}} \right) (1 - \delta) \Pr[\mathcal{E}_i^{(\ell)}] - 2\delta$$

Applying this lower bound to the expression of $L$ and taking the maximum over type 1 units, we have:

$$L \ge 1 + \max_{i \in \mathcal{I}^{(1)}} \left\{ \frac{\mu_{v_i}^{(\tau[1])} - \mu_{v_i}^{(\tau[k])}}{\left( C - 2\alpha(\delta_{PCR}) - 2\sigma\sqrt{\frac{2\log(1/\delta_{\epsilon_i})}{T_1}} \right) (1 - \delta) \Pr\left[ \mathcal{E}_{C,i}^{(\ell)} \right] - 2\delta} \right\}$$

**When unit $i$ is recommended intervention $\tau[1]$:** When unit $i$ gets recommended intervention $\tau[1]$, they know that they are not in the explore group. Hence, the event $\widehat{\mathcal{E}}_{C,i}$ did not happen, and the BIC condition, in this case, can be written as: for any intervention $\tau[j] \ne \tau[1]$,

$$\mathbb{E}[\bar{y}_{i,post}^{(\tau[1])} - \bar{y}_{i,post}^{(\tau[j])} | \neg\widehat{\mathcal{E}}_{C,i}^{(\ell)}] \Pr[\neg\widehat{\mathcal{E}}_{C,i}^{(\ell)}] \ge 0$$

Similar to the previous analysis on the recommendation of intervention $\ell$, it suffices to only consider interventions $\tau[j] < \ell$. We have:

$$\mathbb{E}[\bar{y}_{i,post}^{(\tau[1])} - \bar{y}_{i,post}^{(\tau[j])} | \neg\widehat{\mathcal{E}}_{C,i}^{(\ell)}] \Pr[\neg\widehat{\mathcal{E}}_{C,i}^{(\ell)}] = \mathbb{E}[\bar{y}_{i,post}^{(\tau[1])} - \bar{y}_{i,post}^{(\tau[j])}] - \mathbb{E}[\bar{y}_{i,post}^{(\tau[1])} - \bar{y}_{i,post}^{(\tau[j])} | \widehat{\mathcal{E}}_{C,i}^{(\ell)}] \Pr[\widehat{\mathcal{E}}_{C,i}^{(\ell)}]$$
$$= \mu_{v_i}^{(\tau[1])} - \mu_{v_i}^{(\tau[j])} + \mathbb{E}[\bar{y}_{i,post}^{(\tau[j])} - \bar{y}_{i,post}^{(\tau[1])} | \widehat{\mathcal{E}}_{C,i}^{(\ell)}] \Pr[\widehat{\mathcal{E}}_{C,i}^{(\ell)}]$$

By definition, we have $\mu_{v_i}^{(\tau[1])} \ge \mu_{v_i}^{(\tau[j])}$. Hence, it suffices to show that for any intervention $1 < \tau[j] < \ell$, we have:

$$\mathbb{E}[\bar{y}_{i,post}^{(\tau[j])} - \bar{y}_{i,post}^{(\tau[1])} | \widehat{\mathcal{E}}_{C,i}^{(\ell)}] \Pr[\widehat{\mathcal{E}}_{C,i}^{(\ell)}] \ge 0$$

Observe that with the event $\mathcal{E}$ defined above, we can write:

$$\mathbb{E}[\bar{y}_{i,post}^{(\boldsymbol{\tau}[j])} - \bar{y}_{i,post}^{(\boldsymbol{\tau}[1])}|\widehat{\mathcal{E}}_{C,i}^{(\ell)}]\Pr[\widehat{\mathcal{E}}_{C,i}^{(\ell)}]$$
$$= \mathbb{E}[\bar{y}_{i,post}^{(\boldsymbol{\tau}[j])} - \bar{y}_{i,post}^{(\boldsymbol{\tau}[1])}|\widehat{\mathcal{E}}_{C,i}^{(\ell)}, \mathcal{E}]\Pr[\widehat{\mathcal{E}}_{C,i}^{(\ell)}, \mathcal{E}] + \mathbb{E}[\bar{y}_{i,post}^{(\boldsymbol{\tau}[j])} - \bar{y}_{i,post}^{(\boldsymbol{\tau}[1])}|\widehat{\mathcal{E}}_{C,i}^{(\ell)}, \neg\mathcal{C}]\Pr[\widehat{\mathcal{E}}_{C,i}^{(\ell)}, \neg\mathcal{E}]$$
$$\geq \mathbb{E}[\bar{y}_{i,post}^{(\boldsymbol{\tau}[j])} - \bar{y}_{i,post}^{(\boldsymbol{\tau}[1])}|\widehat{\mathcal{E}}_{C,i}^{(\ell)}, \mathcal{E}]\Pr[\widehat{\mathcal{E}}_{C,i}^{(\ell)}, \mathcal{E}] - 2\delta$$

When event $\widehat{\mathcal{E}}_{C,i}^{(\ell)}$ happens, we know that $\widehat{\mathbb{E}}[\bar{y}_{i,post}^{(\boldsymbol{\tau}[j])}] \geq \widehat{\mathbb{E}}[\bar{y}_{i,post}^{(\boldsymbol{\tau}[1])}] + C$. Furthermore, when event $\mathcal{E}$ happens, we know that $\bar{y}_{i,post}^{(\boldsymbol{\tau}[j])} \geq \widehat{\mathbb{E}}[\bar{y}_{i,post}^{(\boldsymbol{\tau}[j])}] - \alpha(\delta_{PCR}) - \sigma\sqrt{\frac{2\log(1/\delta_{\epsilon_i})}{T_1}}$ and $\widehat{\mathbb{E}}[\bar{y}_{i,post}^{(\boldsymbol{\tau}[1])}] \geq \bar{y}_{i,post}^{(\boldsymbol{\tau}[1])} - \alpha(\delta_{PCR}) - \sigma\sqrt{\frac{2\log(1/\delta_{\epsilon_i})}{T_1}}$. Hence, when these two events $\widehat{\mathcal{E}}_{C,i}^{(\ell)}$ and $\mathcal{E}$ happen simultaneously, we have $\bar{y}_{i,post}^{(\boldsymbol{\tau}[j])} \geq \bar{y}_{i,post}^{(\boldsymbol{\tau}[1])} + C - 2\alpha(\delta_{PCR}) - 2\sigma\sqrt{\frac{2\log(1/\delta_{\epsilon_i})}{T_1}}$. Therefore, the lower bound can be written as:

$$\mathbb{E}[\bar{y}_{i,post}^{(\boldsymbol{\tau}[j])} - \bar{y}_{i,post}^{(\boldsymbol{\tau}[1])}|\widehat{\mathcal{E}}_{C,i}^{(\ell)}]\Pr[\widehat{\mathcal{E}}_{C,i}^{(\ell)}] \geq \left(C - 2\alpha(\delta_{PCR}) - 2\sigma\sqrt{\frac{2\log(1/\delta_{\epsilon_i})}{T_1}}\right)\Pr[\widehat{\mathcal{E}}_{C,i}^{(\ell)}, \mathcal{E}] - 2\delta$$

Similar to the previous analysis when unit $i$ gets recommended intervention $\ell$, we have:

$$\mathbb{E}[\bar{y}_{i,post}^{(\boldsymbol{\tau}[j])} - \bar{y}_{i,post}^{(\boldsymbol{\tau}[1])}|\widehat{\mathcal{E}}_{C,i}]\Pr[\widehat{\mathcal{E}}_{C,i}] \geq \left(C - 2\alpha(\delta_{PCR}) - 2\sigma\sqrt{\frac{2\log(1/\delta_{\epsilon_i})}{T_1}}\right)(1-\delta)\Pr[\mathcal{E}_i^{(\ell)}] - 2\delta$$

Choosing a large enough $C$ such that the right-hand side is non-negative, we conclude the proof. $\qquad\square$

## F   Appendix for Section 5: Testing whether the Unit Overlap Assumption holds

**Theorem 5.2.** *Under Assumption 5.1, if the Unit Overlap Assumption is satisfied for unit $n$, then $|\widehat{\mathbb{E}}[\bar{y}_{n,pre''}] - \bar{y}_{n,pre''}| \leq \alpha(\delta) + 2\sigma\sqrt{\frac{\log(1/\delta)}{T_0}}$ with probability $1 - \mathcal{O}(\delta)$, where $\alpha(\delta)$ is the high-probability confidence interval which is defined in Theorem C.1 when using the first $T_0/2$ time-steps as the pre-intervention period and the next $T_0/2$ time-steps as the post-intervention period.*

*Proof.*

$$|\widehat{\mathbb{E}}[\bar{y}_{i,pre''}] - \bar{y}_{i,pre''}| \leq |\widehat{\mathbb{E}}[\bar{y}_{i,pre''}] - \mathbb{E}[\bar{y}_{i,pre''}]| + |\mathbb{E}[\bar{y}_{i,pre''}] - \bar{y}_{i,pre''}|$$
$$\leq \alpha(\delta) + \left|\frac{2}{T_0}\sum_{t=T_0/2+1}^{T_0}\epsilon_{i,t}^{(0)}\right|$$

with probability at least $1 - \mathcal{O}(\delta)$, where the second inequality follows from Theorem C.1. The result follows from a Hoeffding bound. $\qquad\square$

**Theorem F.1.** *Under Assumption 5.1, if*

$$\frac{r^{3/2}\sqrt{\log(T_0|\mathcal{I}|)}}{\|\widetilde{\omega}^{(n,0)}\|_2 \cdot \min\{T_0, |\mathcal{I}|, T_0^{1/4}|\mathcal{I}|^{1/2}\}} = o(1) \tag{2}$$

*and*

$$\frac{1}{\sqrt{T_1}\|\widetilde{\omega}^{(n,0)}\|_2}\sum_{t=T_0+1}^{T}\sum_{i\in\mathcal{I}}\mathbb{E}[y_{i,t}^{(0)}] \cdot (\widehat{\omega}^{(n,0)}[i] - \widetilde{\omega}^{(n,0)}[i]) = o_p(1) \tag{3}$$

*then as $|\mathcal{I}|, T_0, T_1 \to \infty$ the Asymptotic Hypothesis Test falsely accepts the hypothesis with probability at most 5%, where $\widehat{\sigma}^2$ is defined as in Equation (4) and $\widehat{\omega}^{(n,0)}$ is defined as in Equation (5) in Agarwal et al. (2020b).*

Condition (2) requires that the $\ell_2$ norm of $\widetilde{\omega}^{(n,0)}$ is sufficiently large, and condition (3) requires that the estimation error of $\widehat{\omega}^{(n,0)}$ decreases sufficiently fast. See Section 5.3 of Agarwal et al. (2020b) for more details.

*Proof.* We use $x \xrightarrow{p} y$ (resp. $x \xrightarrow{d} y$) if $x$ converges in probability (resp. distribution) to $y$. Let $\sigma'$ denote the true standard deviation of $\epsilon_{i,t}$. By Theorem 3 of Agarwal et al. (2020b) we know that as $|\mathcal{I}|, T_0, T_1 \to \infty$,

1. $\frac{\sqrt{T_1}}{\sigma' \|\widetilde{\omega}^{(n,0)}\|_2} (\widehat{\mathbb{E}}[\bar{y}_{n,pre''}] - \mathbb{E}[\bar{y}_{n,pre''}]) \xrightarrow{d} \mathcal{N}(0,1)$

2. $\widehat{\sigma} \xrightarrow{p} \sigma'$ and

3. $\widehat{\omega}^{(n,0)} \xrightarrow{p} \widetilde{\omega}^{(n,0)}$.

By the continuous mapping theorem, we know that $\frac{1}{\widehat{\sigma}} \xrightarrow{p} \frac{1}{\sigma'}$ and $\frac{1}{\|\widehat{\omega}^{(n,0)}\|_2} \xrightarrow{p} \frac{1}{\|\widetilde{\omega}^{(n,0)}\|_2}$. We can rewrite $\widehat{\mathbb{E}}[\bar{y}_{n,pre''}] - \mathbb{E}[\bar{y}_{n,pre''}]$ as $\widehat{\mathbb{E}}[\bar{y}_{n,pre''}] - \bar{y}_{n,pre''} + \bar{\epsilon}_{n,pre''}$, and we know that $\bar{\epsilon}_{n,pre''} \xrightarrow{p} 0$. Applying Slutksy's theorem several times obtains the desired result. $\square$

## G    Appendix for Section 6: Numerical simulations

**Experimental Setup.** If unit $i$ is of type 1 (resp. type 0), we generate a latent factor $\mathbf{v}_i = [0 \ \cdots \ 0 \ v_i[1] \ \cdots \ v_i[r/2]]$ (resp. $\mathbf{v}_i = [v_i[1] \ \cdots \ v_i[r/2] \ 0 \ \cdots \ 0]$), where $\forall j \in [r/2] : v_i[j] \sim \mathsf{Unif}(0,1)$. We consider an online setting where 500 units of alternating types arrive sequentially.[12] We consider a pre-intervention time period of length $T_0 = 100$ with latent factors as follows:

- If $t \mod 2 = 0$, generate latent factor
  $\mathbf{u}_t^{(0)} = [0 \cdots 0 \ u_t^{(0)}[1] \cdots u_t^{(0)}[r/2]]$, where $\forall \ell \in [r/2] : u_t^{(0)}[\ell] \sim \mathsf{Unif}[0.25, 0.75]$.

- If $t \mod 2 = 1$, generate latent factor
  $\mathbf{u}_t^{(0)} = [u_t^{(0)}[1] \cdots u_t^{(0)}[r/2] \ 0 \cdots 0]$, where $\forall \ell \in [r/2] : u_t^{(0)}[\ell] \sim \mathsf{Unif}[0.25, 0.75]$.

We consider a post-intervention period of length $T_1 = 100$ and generate the following post-intervention latent factors:

- If $d = 0$, we set $\mathbf{u}_t^{(d)} = [u_t^{(d)}[1] \ \cdots \ u_t^{(d)}[r]] : t \in [T_0 + 1, T]$, where $\forall \ell \in [r] : u_t^{(d)}[\ell] \sim \mathsf{Unif}[0,1]$.

- If $d = 1$, we set $\mathbf{u}_t^{(d)} = [u_t^{(d)}[1] \ \cdots \ u_t^{(d)}[r]] : t \in [T_0 + 1, T]$, where $\forall \ell \in [r] : u_t^{(d)}[\ell] \sim \mathsf{Unif}[-1,0]$.

Finally, outcomes are generated by adding independent Gaussian noise $\epsilon_{i,t}^{(d)} \sim \mathcal{N}(0, 0.01)$ to each inner product of latent factors.

**Non-compliance setting**    Following the standard in the literature on IE, we assume that all units are Bayesian-rational (Definition 2.7). That is, given a recommendation $\widehat{d}_i$, unit $i$ will select an intervention $d_i$ that maximizes their utility in expectation over their prior conditional on the recommendation $\widehat{d}_i$. However, in practice, units may exhibit sub-rational behaviors, where they neglect the principal's recommendation and act according to their initial prior. In Figure 4, we examine the performance of Algorithm 1 in the presence of such non-compliance behavior.

We use the same data-generating process as described above. When running Algorithm 1, we let a random $p-$proportion of the units ignore the principal's recommendation and choose their initially preferred intervention, i.e., type 1 units will select the treatment and type 0 units will select the control. In Figure 4, we vary the non-compliance parameter $p \in \{0, 0.2, 0.4, 0.6, 0.8\}$ and calculate the counterfactual estimation error for units of type 1 under control. As the non-compliance level $p$ increases, it becomes more challenging to persuade units to explore, and the counterfactual estimation error increases.

---

[12]The alternating arrival of different unit types is done only for convenience. The order of unit arrival is unknown to the algorithms.

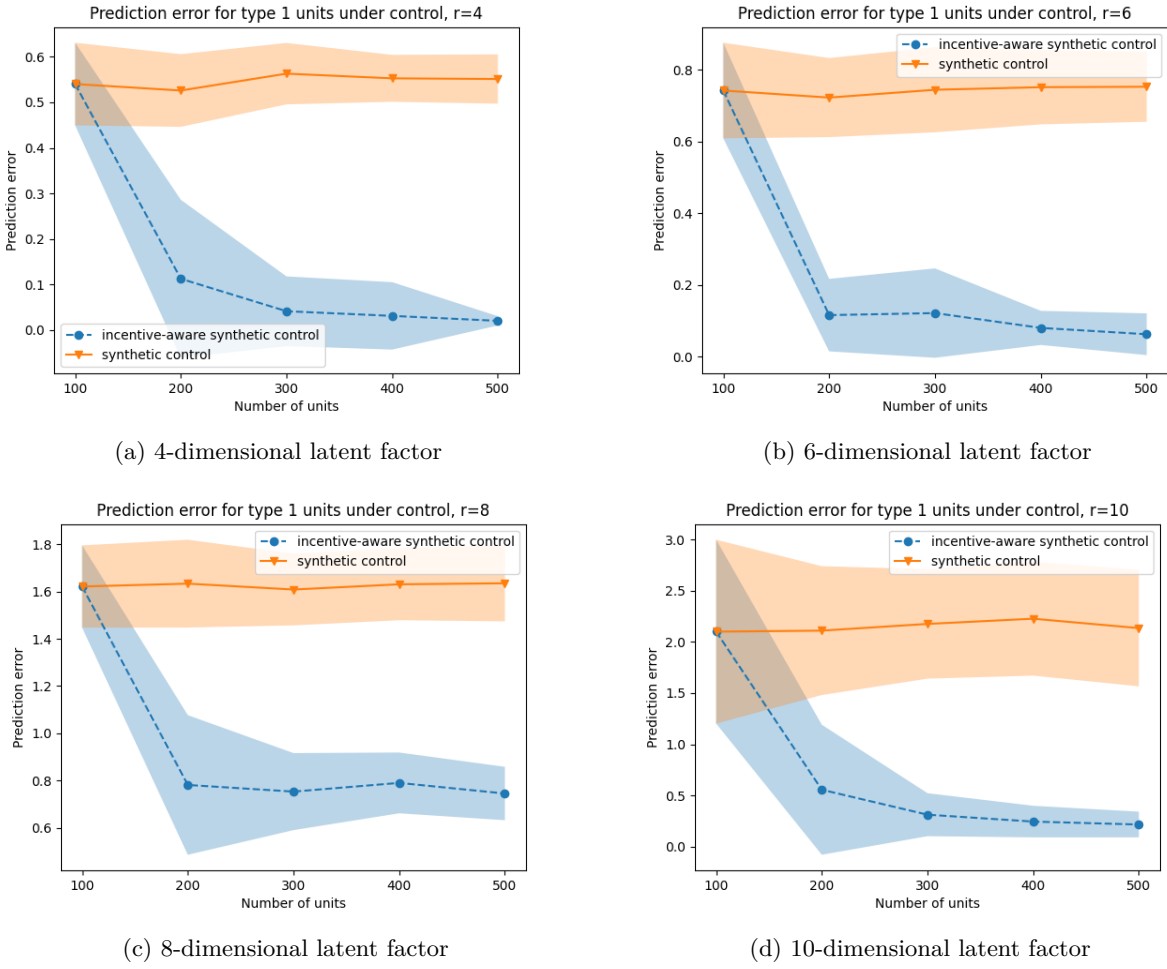

Figure 3: Counterfactual estimation error for units of type 1 under control using Algorithm 1 (blue) and synthetic control without incentives (orange) with increasing number of units and different lengths of the latent factors. Results are averaged over 50 runs, with the shaded regions representing one standard deviation.

**Violation of Linear Span Inclusion assumption 3.2**  Similar to prior work in the literature on synthetic control, our theoretical framework relies on the linear span inclusion assumption (Assumption 3.2), which is necessary to ensure the counterfactual estimation procedure in Appendix B is accurate. In Figure 5, we investigate the performance of our algorithm when this assumption does not hold.

Specifically, we make the following modifications to the data-generating process. If unit $i$ is of type 1 (resp. type 0), we generate a latent factor $\mathbf{v}_i = [0 \; \cdots \; 0 \; v_i[1] \; \cdots \; v_i[(r-1)/2] \; 1]$ (resp. $\mathbf{v}_i = [v_i[1] \; \cdots \; v_i[(r-1)/2] \; 0 \; \cdots \; 0 \; 1]$), where $\forall j \in [(r-1)/2] : v_i[j] \sim \mathsf{Unif}(0,1)$. In the pre-intervention time period of length $T_0 = 100$, the time latent factors are generated with a padded entry 0 at the end as follows:

- If $t \mod 2 = 0$, generate latent factor $\mathbf{u}_t^{(0)} = [\underbrace{0 \cdots 0}_{(r-1)/2} \; \underbrace{u_t^{(0)}[1] \cdots u_t^{(0)}[(r-1)/2]}_{(r-1)/2} \; 0]$, where $\forall \ell \in [(r-1)/2] :$

  $u_t^{(0)}[\ell] \sim \mathsf{Unif}[0.25, 0.75]$.

- If $t \mod 2 = 1$, generate latent factor $\mathbf{u}_t^{(0)} = [\underbrace{u_t^{(0)}[1] \cdots u_t^{(0)}[(r-1)/2]}_{(r-1)/2} \; \underbrace{0 \cdots 0}_{(r-1)/2} \; 0]$, where $\forall \ell \in [(r-1)/2] :$

  $u_t^{(0)}[\ell] \sim \mathsf{Unif}[0.25, 0.75]$.

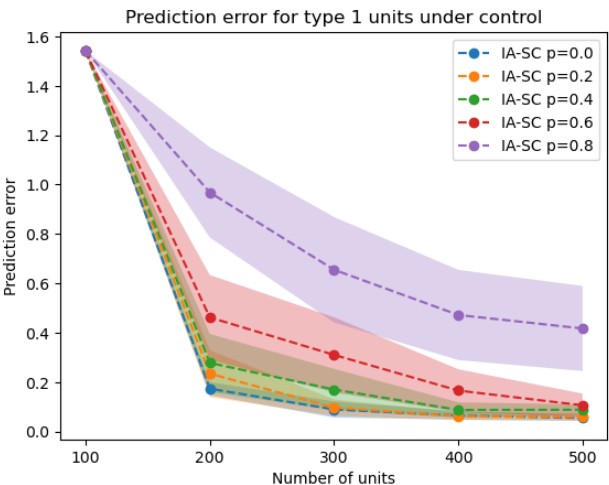

Figure 4: Counterfactual estimation error for units of type 1 under control using Algorithm 1 with increasing number of units and different non-compliance level $p \in \{0.0, 0.2, 0.4, 0.6, 0.8\}$. Results are averaged over 50 runs, with the shaded regions representing one standard error.

In the post-intervention period of length $T_1 = 100$, the time-and-intervention latent factors are generated as follows.

- If $d = 0$, with probability $p$, we set $\mathbf{u}_t^{(d)} = [u_t^{(d)}[1] \ \cdots \ u_t^{(d)}[r]] : t \in [T_0 + 1, T]$, where $\forall \ell \in [r] : u_t^{(d)}[\ell] \sim$ $\mathsf{Unif}[0, 1]$; and with probability $1 - p$, we set $\mathbf{u}_t^{(d)} = [u_t^{(d)}[1] \ \cdots \ u_t^{(d)}[r - 1] \ 0] : t \in [T_0 + 1, T]$, where $\forall \ell \in [r - 1] : u_t^{(d)}[\ell] \sim \mathsf{Unif}[0, 1]$.

- If $d = 1$, with probability $p$, we set $\mathbf{u}_t^{(d)} = [u_t^{(d)}[1] \ \cdots \ u_t^{(d)}[r]] : t \in [T_0 + 1, T]$, where $\forall \ell \in [r] : u_t^{(d)}[\ell] \sim$ $\mathsf{Unif}[-1, 0]$; with probability $1 - p$, we set $\mathbf{u}_t^{(d)} = [u_t^{(d)}[1] \ \cdots \ u_t^{(d)}[r - 1] \ 0] : t \in [T_0 + 1, T]$, where $\forall \ell \in [r - 1] : u_t^{(d)}[\ell] \sim \mathsf{Unif}[-1, 0]$.

That is, with probability $p$, the post-intervention latent factor $\mathbf{u}_t^{(d)}$ does not lie in the span of the pre-intervention latent factors $\{\mathbf{u}_1^{(0)}, \cdots \mathbf{u}_{T_0}^{(0)}\}$. In Figure 5, as the probability $p$ of post-intervention latent factors not lying in the pre-intervention span increases, i.e., more severe violation of Assumption 3.2, the counterfactual estimation error does not decrease to zero and instead may increase as more units arrive.

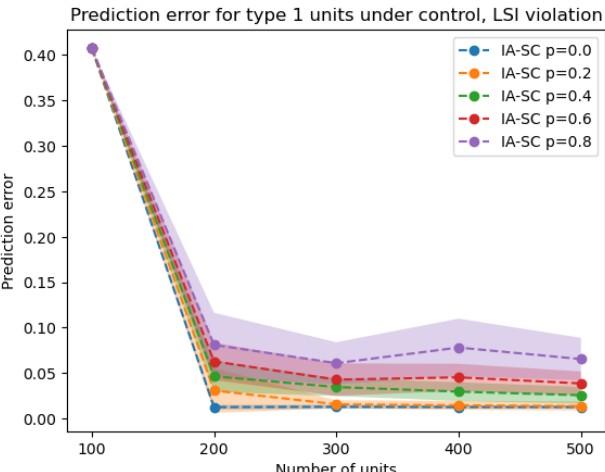

Figure 5: Counterfactual estimation error for units of type 1 under control using Algorithm 1 with increasing number of units and probability of linear span inclusion violation $p \in \{0, 0.2, 0.4, 0.6, 0.8\}$. Results are averaged over 50 runs, with the shaded regions representing one standard error.

