# OpenReview forum: "Incentive-Aware Synthetic Control: Accurate Counterfactual Estimation via Incentivized Exploration"
_TMLR — Accepted by TMLR_

### Review · Reviewer_xHZa · 2025-12-01

**Summary Of Contributions:**

This paper studies synthetic control methods in settings where units self-select into interventions, so the usual “unit overlap” assumption may systematically fail. The authors formally show that without overlap, SCMs cannot recover certain counterfactuals even with infinite data, propose an incentive-aware exploration policy that actively creates overlap while remaining Bayesian incentive-compatible, and provide diagnostics to test for overlap plus simulations that illustrate both the failure of standard SCM and the benefits of their approach.

The main contributions are:
- Demonstrating the necessity of unit overlap for SCM.
- Introducing an incentive-aware exploration mechanism to restore overlap.
- Providing practical diagnostics for overlap and simulation-based evidence.

# Strengths
- Clarifies a fundamental limitation of SCM in self-selection settings.
By proving that unit overlap is not just convenient but necessary for consistent SCM estimation, the paper sharpens our understanding of when SCM can and cannot be trusted. This is especially valuable because SCM is often used in observational contexts where self-selection is strong, yet the overlap assumption is rarely interrogated. The explicit counterexample with non-overlapping latent types is simple but powerful: it provides a concrete mental model practitioners can use to reason about whether their own setting might violate overlap.
- Innovative integration of causal inference and incentive design.
The work brings together synthetic control, latent factor models, and incentive-compatible exploration. Rather than trying to “fix” SCM post hoc, it changes the data-collection process via recommendations so that SCM’s assumptions become true over time. This is conceptually elegant: the platform uses early data to learn which units are likely to belong to different latent types, then deliberately sends some of them to “unlikely” interventions in a way that still respects their incentives. The mechanism design angle is non-standard in SCM and opens a promising direction: designing recommendation policies that are both user-aligned and statistically informative.

- Solid theoretical development with clear logic and connections to prior work.
The paper’s main theorems are built on a clear latent factor framework and make sensible use of known results from robust SCM and PCR. The move from vertical SCM to a horizontal regression view is particularly helpful: it simplifies the online setting and links the problem to contextual bandit intuition. Overall, the theoretical narrative is coherent: it starts from a stark impossibility, then shows how changing incentives and recommendations can get around it.

- Useful diagnostics via overlap testing.
The two proposed tests for unit overlap give practitioners a practical handle on a condition that is otherwise quite abstract. In SCM applications today, overlap is often assumed without being checked. The paper’s tests provide a way to monitor whether individual units are in- or out-of-support for a given donor pool. Even aside from the specific incentive mechanism, these diagnostics could be adopted in existing SCM workflows as a pre-check or sensitivity tool to flag units whose counterfactuals are inherently hard to identify.

- Synthetic experiments that closely match the theoretical story.
The simulations are carefully designed to instantiate the latent-type, non-overlapping-subspace structure that motivates the theory and to mimic the online arrival of units under self-selection. As a result, the empirical plots directly reflect the claims: baseline SCM stalls at a nonzero error, whereas the incentive-aware algorithm drives error down; the overlap tests behave as predicted by the theorems. This tight theory–experiment alignment makes it easy for readers to see the mechanisms at work and trust that the formal results are not just artifacts of overfitting a proof.

# Weaknesses
- Strong and somewhat opaque assumptions for incentive compatibility and parameter tuning.
The guarantees for the main algorithm hinge on technical assumptions and knowledge of various bounds (on latent factors, noise, etc.) that both the platform and the agents are assumed to share. In the current presentation, these requirements are mostly relegated to the appendix and not fully unpacked in the main text. For a reader thinking about deployment, it is not obvious how realistic it is to know or estimate these quantities, nor how sensitive BIC and overlap guarantees are to misspecification. This makes the theoretical results feel somewhat “brittle”: mathematically clean, but with unclear robustness when the assumed bounds are only roughly known.

- Idealized behavioral model for users.
The BIC notion presumes users are fully Bayesian, know the recommendation policy, and condition correctly on receiving a recommendation when deciding whether to follow it. Real users on platforms may be myopic, heterogeneous in sophistication, or simply not pay close attention to recommendation logic. The paper does not explore how much deviation from the idealized BIC setting the mechanism can tolerate while still creating enough exploration to restore overlap. As a result, there is a gap between the theoretical guarantees (which are very clean) and the behavioral complexity of actual recommendation or pricing environments.

- Narrow structural model and limited exploration of model misspecification.
The core intuition is built around units belonging to distinct latent “types” with non-overlapping subspaces under different interventions. While this is a compelling stylized model, the paper does relatively little to examine more nuanced situations: partly overlapping subspaces, many types, or smoother heterogeneity structures. Similarly, the simulations closely mirror the assumed latent factor structure, so we don’t see how the method behaves when the model is mis-specified (e.g., true dynamics are nonlinear, or the latent rank is under- or over-estimated). This limits our ability to judge how robust the approach is outside of the carefully crafted theoretical regime.

- Empirical evaluation confined to synthetic data.
All empirical evidence is based on synthetic panels designed to match the theory. While this is appropriate for validating the math, it means the paper does not demonstrate its methods on any real or semi-synthetic SCM benchmarks (e.g., policy interventions, marketing campaigns, pricing experiments). Without at least a small real-world or semi-synthetic example, it’s hard to assess issues like data sparsity, unmodeled covariates, or noisy behavioral responses that often arise in practice. This weakens the case for immediate practical impact, even though the conceptual contribution is strong.

**Audience:**

Yes

**Audience Explanation:**

The work is highly relevant to the SCM and causal inference community and online learning community.

**Claims And Evidence:**

Yes

**Claims Explanation:**

The theoretical claims are well supported within their modeling framework, and the simulations convincingly illustrate the main phenomena under those assumptions, but the empirical backing for real-world robustness is limited.

**Requested Changes:**

1. Currently Theorem 4.1 refers to Assumption C.3 with little intuition in the main text. Could you summarize in the main body:
- What exactly does Assumption C.3 require (in qualitative terms)?
- Which moments/parameters of the population must the principal and units know (or upper/lower bound)?
- Why is it reasonable in intended applications?

Similarly, could you give a high-level description of how $N_0$, $L$, $B$ in Alg.1 are chosen from observable quantities, with a concrete example or at least an order-of-magnitude sketch.

2. Add more discussion of behavioral assumptions and BIC interpretation. The BIC argument assumes Bayesian-rational agents who know the algorithm and can condition correctly on receiving a given recommendation. Could you add a dedicated subsection discussing:

- To what extent this aligns with realistic user behavior in systems like recommender or pricing systems.
- Whether “approximate” BIC (users loosely trusting recommendations) suffices for the algorithm to still produce overlap.
- Any robustness to misspecified beliefs or heterogeneous sophistication.

3. It would be good to have an experiment that probes robustness beyond the idealized model. Without necessarily requiring a full real-world dataset, it would be valuable to have a simulation that breaks some assumptions in a controlled way, e.g.:
- Latent subspaces for types only partially orthogonal.
- Some units misfollowing recommendations (e.g., follow with probability <1).
- Mild violations of linear span inclusion.

and report whether the algorithm still substantially improves estimation relative to baseline SCM.

---

> ### Author Response · Authors · 2026-01-04
>
> We thank the reviewer for their thoughtful comments and questions. Please find our replies to your comments below.
>
> Regarding the assumptions we make (Assumption C.3), we would like to emphasize that they are, by and large, assumptions that the principal and units know valid upper- and lower-bounds on various population-level statistics (as opposed to assumptions on exact quantities on user-level statistics, which we do not assume). As a result, the principal may have general knowledge of these bounds or be able to learn them by, e.g., running studies on a subset of the population. Similarly, the units may have a general knowledge about the overall population they belong to, or the principal may choose to make this information publicly available. We would also like to mention that such assumptions are quite common (and are often necessary) in the literature on incentivized exploration.
>
> We agree that examining the impact of non-Bayesian agents is an interesting direction to explore. As a step in this direction, we have added additional experiments in Appendix G, which show the performance of our methods when some fraction of the agent population does not follow the recommendations. We find that our performance degrades gracefully as this fraction of non-followers increases. We leave further study of non-Bayesian agents as a direction for future work.
>
> Regarding model misspecification, we have conducted additional experiments in which a fraction of the post-intervention latent factors violate the linear span inclusion assumption. As was the case above, we find that our performance degrades gracefully as a function of the fraction of latent factors that do not satisfy this condition. All details for this experiment are included in Appendix G.
>
> Regarding “partially orthogonal subspaces”, we would like to clarify that if there is partial overlap between the latent subspaces for the different types, then the unit overlap assumption will eventually be satisfied on its own, given enough samples from the underlying agent population. With that being said, it may be possible for the principal to speed up this process by using our method.
>
> Regarding a high-level description/example of how the parameters of Algorithm 1 are chosen from observable quantities, we actually have already provided such an example in the appendix (Example C.6). We chose to include it in the appendix instead of the main body due to length considerations and so as not to interrupt the flow of the paper. However, we explicitly reference this example in the last sentence of Section 4.
>
> Regarding all other requested changes to the paper’s writing, we have implemented them to the best of our ability in the revised manuscript. However, please let us know if there is anything you feel we did not adequately address.

---

### Review · Reviewer_gKLn · 2025-12-11

**Summary Of Contributions:**

The paper offers novel contributions to counterfactual causal estimation by bringing ideas from online active interventions. Most importantly, in my opinion, is that it provides an impossibility result showing that without unit overlap, no estimator can consistently recover counterfactual outcomes in latent-factor panel models with self-selected interventions. Motivated by these observations, the paper introduces an incentive-based exploration control that encourages units to try interventions they would not usually choose. This promotes creating intervention overlaps. This is a departure from prior SCM studies, which assume the experimenter controls interventions and do not model strategic agents.

**Audience:**

Yes

**Audience Explanation:**

It will be of interest to those working on causality, especially in the potential outcome framework.

**Broader Impact Concerns:**

There is no broader statement section. Nevertheless, I do not notice any ethical implications.

**Claims And Evidence:**

Yes

**Claims Explanation:**

Overall, the contribution is well-motivated and technically sound, and I would like to recommend it for publication. Before that, however, there are a couple of issues that I invite the authors to consider for revising the manuscript.

1-- The key ideas about how online explorations are induced and the two-step intervention strategy are primarily presented in the introduction. However, most of the statements are not written clearly enough. This took me a few times reading the paper to fully understand some of the statements fully. For instance, the second paragraph of the introduction has a key role, and the choice of words and references in the description of the two stages is not clear enough.

2-- The scope of the paper has some indirect, but important, connections to contextual bandits, linear bandits, and causal bandits. I recommend that the authors include a discussion of connections to the relevant literature, especially recent advances in causal bandits.

3-- The streaming service example could be an interesting case for identifying gaps in the literature. However, as written, the message is unclear. I encourage the authors to rewrite it more in line with the key elements of the model (i.e., how pre-intervention, intervention, and post-intervention phases manifest in this example).

4-- The paper argues that the Unit Overlap Assumption is necessary for consistent counterfactual estimation. While the constructed example seems correct, the authors should more explicitly position this result within identifiability discussions in generalized SCM, existing overlap/common-support assumptions used in SCM inference papers, impossibility results in causal inference, and panel identification. Currently, the claims are as if SCM literature has entirely ignored the necessity of overlap, whereas several papers mention related conditions, though not with this formal impossibility structure. Clarifying what is completely novel and what the result adds to prior insights would improve the contribution's credibility.

5-- The presentation of the extension to more than two interventions is a bit too busy and complex. I suggest that the authors provide a discussion on why the ordering structure (sub-types) is natural, when such a structure arises in real applications, and how exploration interacts across multiple treatments in a more intuitive way.

**Requested Changes:**

Please refer to comments 1--5 earlier. They all provide recommendations for changes and modifications.

---

> ### Author Response · Authors · 2026-01-04
>
> Thank you for your detailed suggestions on how to improve our paper! We have addressed your comments to the best of our abilities in the revised manuscript, but please let us know if there is any part that you feel is not sufficient.
>
> Regarding (1), we expanded on the second paragraph in the introduction and reworded a few things for clarity.
>
> Regarding (2), we added a new paragraph on the conceptual connections to causal bandits in the additional related work section (Appendix A).
>
> Regarding (3), we expanded the paragraph on the streaming example to clarify which parts of the example correspond to the synthetic control setup.
>
> Regarding (4), we added a paragraph in the related work (Section 1.1) on the unit overlap assumption and related conditions in the literature.
>
> Finally, regarding (5), we have added a paragraph explaining the relationship between incentivizing exploration for multiple sub-types (Appendix D). We would like to clarify that it is without loss of generality to assume that units have a preference ordering over the interventions (as long as tie-breaking is done in a well-defined and deterministic way). A unit's subtype is just this ordering over interventions. Since there are k! possible orderings of the k interventions, there are k! possible subtypes.

---

### Review · Reviewer_zhae · 2025-12-25

**Summary Of Contributions:**

This paper studies counterfactual inference with panel data when treatment assignment is endogenous: units have heterogeneous preferences over interventions and self-select, so the usual synthetic control identification condition (“unit overlap”) can fail exactly in the cases of practical interest.  The first main contribution is a sharp necessity result: if the Unit Overlap Assumption fails for the target unit/intervention, then no algorithm can consistently recover the counterfactual (Theorem 3.1 provides an explicit impossibility construction).  The second contribution is methodological: the authors design an “incentive-aware synthetic control” recommendation policy that creates overlap by persuading some units to take otherwise suboptimal interventions, adapting the “hidden exploration” paradigm from incentivized exploration so that recommendations remain Bayesian incentive-compatible.  They show that after interacting with sufficiently many units, overlap holds with high probability, enabling valid counterfactual estimation via off-the-shelf robust SCM/PCR guarantees.  Finally, the paper adds practical diagnostics (two hypothesis tests for overlap) and extends the approach to multiple interventions via synthetic interventions, supported by simulation evidence.

**Audience:**

Yes

**Audience Explanation:**

A meaningful subset of TMLR’s audience in online learning and sequential experimentation should find this paper relevant. The main conceptual move is to recast a canonical panel-data counterfactual task (synthetic control) as a problem of data acquisition under endogenous participation: when users self-select among interventions, the platform may never observe the variation needed for reliable counterfactual estimation. The paper’s response—an incentive-aware recommendation policy that implements “hidden exploration” while maintaining Bayesian incentive compatibility—connects synthetic control to core themes in bandits and incentivized exploration: how to design exploration policies when feedback is filtered through strategic or preference-driven behavior.

The paper also speaks to recommender-systems and platform readers by framing counterfactual identification as a platform design issue rather than a purely statistical one. In settings where interventions correspond to product choices, plans, or policies offered to heterogeneous users, the paper argues that purely observational data can be non-overlapping precisely because the recommendation environment is already optimized for user utility. The proposed mechanism provides a principled way to inject controlled exploration via recommendations so that the platform can learn counterfactual outcomes, making the contribution legible to “internet economy” contexts where learning and incentive alignment are inseparable.

**Broader Impact Concerns:**

None.

**Claims And Evidence:**

Yes

**Claims Explanation:**

In fact, due to my lack of expertise in synthetic control method, I'm unable to fully understand and verify the correctness of the results. The writing does look sound at a glance.

**Requested Changes:**

The authors should try to rewrite the paper to be more accessible for readers without background knowledge about synthetic control.

---

> ### Author Response · Authors · 2026-01-04
>
> We thank the reviewer for their thoughtful comments. We have made several changes to the writing in the revised manuscript, which we hope will make the paper more accessible for readers without previous background knowledge in synthetic control methods. In particular, we have separated out the mathematical description of our synthetic control setup into its own subsection (Section 2.1), and added a forward pointer to this section in the introduction. We expanded on the second paragraph in the introduction and reworded a few things for clarity. We have also fleshed out our streaming example in the introduction to clarify which parts of the example correspond to the synthetic control setup. In Section 4, we added a high-level description of some of our technical assumptions. Finally, we added a paragraph in the related work section (Section 1.1) on the unit overlap assumption and related conditions in the literature, and we added a paragraph in the additional related work section (Appendix A) on the connections to causal bandits.

---

### Decision · Action_Editor_8Xy3 · 2026-02-24

**Recommendation:** Accept as is

**Audience:**

Yes

**Audience Explanation:**

Yes. It should interest multiple ML subcommunities: researchers in causal inference and synthetic control (new approach to overlap/identifiability under self-selection), bandits and incentivized exploration (BIC recommendation design and hidden-explore batching), and applied ML/recommender-system teams that care about estimation under strategic users.

**Claims And Evidence:**

Yes

**Claims Explanation:**

The paper identifies a common failure mode for synthetic control methods: when units self-select interventions, subpopulations can be non-overlapping so donor-based counterfactuals are invalid; to fix this it designs incentive-compatible recommendation policies that induce targeted exploration (hidden-explore batching algorithms: Algorithm 1 for two arms and Algorithm 2 for k arms) so that, with calibrated batch sizes and an initial no-recommendation phase, the Unit Overlap Assumption holds with high probability. The claims in the paper are assessed to be reasonable.